# Enhancing Graph Transformers with Hierarchical Distance Structural Encoding

Yuankai Luo[1,3]        Hongkang Li[2]        Lei Shi[1]*        Xiao-Ming Wu[3]*

[1]Beihang University        [2]Rensselaer Polytechnic Institute

[3]The Hong Kong Polytechnic University

luoyk@buaa.edu.cn            xiao-ming.wu@polyu.edu.hk

## Abstract

Graph transformers need strong inductive biases to derive meaningful attention scores. Yet, current methods often fall short in capturing longer ranges, hierarchical structures, or community structures, which are common in various graphs such as molecules, social networks, and citation networks. This paper presents a Hierarchical Distance Structural Encoding (HDSE) method to model node distances in a graph, focusing on its multi-level, hierarchical nature. We introduce a novel framework to seamlessly integrate HDSE into the attention mechanism of existing graph transformers, allowing for simultaneous application with other positional encodings. To apply graph transformers with HDSE to large-scale graphs, we further propose a high-level HDSE that effectively biases the linear transformers towards graph hierarchies. We theoretically prove the superiority of HDSE in terms of expressivity and generalization. Empirically, we demonstrate that graph transformers with HDSE excel in graph classification, regression on 7 graph-level datasets, and node classification on 11 large-scale graphs.

## 1 Introduction

The success of Transformers [74] in various domains, including natural language processing (NLP) and computer vision [18], has sparked significant interest in developing transformers for graph data [19, 86, 46, 11, 68, 61, 89, 60, 82]. Scholars have turned their attention to this area, aiming to address the limitations of Message-Passing Graph Neural Networks (MPNNs) [30] such as over-smoothing [54] and over-squashing [1, 73].

However, Transformers [74] are known for their lack of strong inductive biases [18]. In contrast to MPNNs, graph transformers do not rely on fixed graph structure information. Instead, they compute pairwise interactions for all nodes within a graph and represent positional and structural data using more flexible, soft inductive biases. Despite its potential, this mechanism does not have the capability to learn hierarchical structures within graphs. Developing effective positional encodings is also challenging, as it requires identifying important hierarchical structures among nodes, which differ significantly from other Euclidean domains [10]. Consequently, graph transformers are prone to overfitting and often underperform MPNNs when data is limited [61], especially in tasks involving large graphs with relatively few labeled nodes [82]. These challenges become even more significant when dealing with various molecular graphs, such as those found in polymers or proteins. These graphs are characterized by a multitude of substructures and exhibit long-range and hierarchical structures. The inability of graph transformers to learn these hierarchical structures can significantly impede their performance in tasks involving such complex molecular graphs.

---

*Corresponding authors.

38th Conference on Neural Information Processing Systems (NeurIPS 2024).

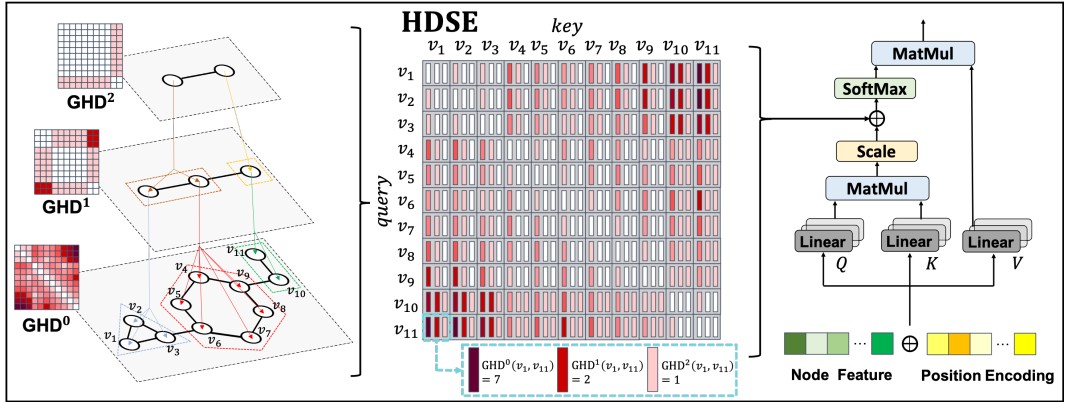

Figure 1: Overview of our proposed hierarchical distance structural encoding (HDSE) and its integration with graph transformers. HDSE uses the graph hierarchy distance (GHD, refer to Definition 1) that can capture interpretable patterns in graph-structured data by using diverse graph coarsening algorithms. Darker colors indicate longer distances.

Further, the global all-pair attention mechanism in transformers poses a significant challenge due to its time and space complexity, which increases quadratically with the number of nodes. This quadratic complexity significantly restricts the application of graph transformers to large graphs, as training them on graphs with millions of nodes can require substantial computational resources. Large-scale graphs, such as social networks and citation networks, often exhibit community structures characterized by closely interconnected groups with distinct hierarchical properties. To enhance the scalability and effectiveness of graph transformers, it is crucial to incorporate hierarchical structural information at various levels.

To address the aforementioned challenges and unlock the true potential of the transformer architecture in graph learning, we propose a Hierarchical Distance Structural Encoding (HDSE) method (**Sec. 3.1**), which can be combined with various graph transformers to produce more expressive node embeddings. HDSE encodes the hierarchy distance, a metric that measures the distance between nodes in a graph, taking into account multi-level graph hierarchical structures. We utilize popular coarsening methods [43, 65, 31, 6, 57] to construct graph hierarchies, enabling us to measure the distance relationship between nodes across various hierarchical levels.

HDSE enables us to incorporate a robust inductive bias into existing transformers and address the issue of lacking canonical positioning. To achieve this, we introduce a novel framework (**Sec. 3.2**), as illustrated in Figure 1. We utilize an end-to-end trainable function to encode HDSE as structural bias weights into the attentions, allowing the graph transformer to integrate both HDSE and other positional encodings simultaneously. Our theoretical analysis demonstrates that *graph transformers equipped with HDSE are significantly more powerful than the ones with the commonly used shortest path distances or without relative positional encodings, in terms of both expressiveness and generalization*. We rigorously evaluate our HDSE in ablation studies and show that it successfully improves different kinds of baseline transformers, from vanilla graph transformers [19] to state-of-the-art graph transformers [68, 11, 91, 61], across 7 graph-level datasets.

To enable the application of graph transformers with HDSE to large graphs ranging from millions to billions of nodes, we introduce a high-level HDSE (**Sec. 3.3**), which effectively biases the linear transformers towards the multi-level structural nature of these large networks. We demonstrate our high-level HDSE method exhibits high efficiency and quality across 11 large-scale node classification datasets, with sizes up to the billion-node level.

Our implementation is available at https://github.com/LUOyk1999/HDSE.

## 2 Background and Related Works

We refer to a *graph* as a tuple $G = (V, E, \mathbf{X})$, with node set $V$, edge set $E \subseteq V \times V$, and node features $\mathbf{X} \in \mathbb{R}^{|V| \times d}$. Each row in $\mathbf{X}$ represents the feature vector of a node, with $|V|$ denoting the number of nodes and feature dimension $d$. The features of node $v$ are denoted by $x_v \in \mathbb{R}^d$.

Table 1: Comparison of popular graph coarsening algorithms.

| Coarsening algorithm | METIS [43] | Spectral [65] | Loukas [57] | Newman [31] | Louvain [6] |
|:---:|:---:|:---:|:---:|:---:|:---:|
| Complexity | $O(|E|)$ | $O(|V|^3)$ | $O(|V|)$ | $O(|E|^2|V|)$ | $O(|V|\log|V|)$ |

## 2.1 Graph Hierarchies

Given an input graph $G$, a graph hierarchy of $G$ consists of a sequence of graphs $(G^k, \phi_k)_{k \geq 0}$, where $G^0 = G$ and $\phi_k : V^k \to V^{k+1}$ are surjective node mapping functions. Each node $v_j^{k+1} \in V^{k+1}$ represents a *cluster* of a subset of nodes $\{v_i^k\} \subseteq V^k$. This partition can be described by a projection matrix $\hat{P}^k \in \{0,1\}^{|V^k| \times |V^{k+1}|}$, where $\hat{P}_{ij}^k = 1$ if and only if $v_j^{k+1} = \phi_k(v_i^k)$. The normalized version can be defined by $P^k = \hat{P}^k C^{k-1/2}$, where $C^k \in \mathbb{R}^{|V^{k+1}| \times |V^{k+1}|}$ is a diagonal matrix with its $j$-th diagonal entry being the cluster size of $v_j^{k+1}$. We define the node feature matrix $\mathbf{X}^{k+1}$ for $G^{k+1}$ by $\mathbf{X}^{k+1} = P^{k\top} \mathbf{X}^k$, where each row of $\mathbf{X}^{k+1}$ represents the average of all entries within a specific *cluster*. The edge set $E^{k+1}$ of $G^{k+1}$ is defined as $E^{k+1} = \{(u^{k+1}, v^{k+1}) | \exists v_r^k \in \phi_k^{-1}(u^{k+1}), v_s^k \in \phi_k^{-1}(v^{k+1}), \text{ such that } (v_r^k, v_s^k) \in E^k\}$.

Graph hierarchies can be constructed by repeatedly applying graph coarsening algorithms. These algorithms take a graph, $G^k$, and generate a mapping function $\phi_k : V^k \to V^{k+1}$, which maps the nodes in $G^k$ to the nodes in the coarser graph $G^{k+1}$. A summary and comparison of popular graph coarsening algorithms, along with their computational complexities, can be found in Table 1. We define the *coarsening ratio* as $\alpha = \frac{|V^{k+1}|}{|V^k|}$, which represents the proportion of the number of nodes in the coarser graph $G^{k+1}$ to the number of nodes in the original graph $G^k$. Consequently, each graph $G^k$, where $k > 0$, captures specific substructures derived from the preceding graph.

## 2.2 Graph Transformers

Transformers [74] have recently gained significant attention in graph learning, due to their ability to learn intricate relationships that extend beyond the capabilities of conventional GNNs [44, 33, 59, 58], and in a unique way. The architecture of these models primarily contain a *self-attention* module and a feed-forward neural network, each of which is followed by a residual connection paired with a normalization layer. Formally, the self-attention mechanism involves projecting the input node features $\mathbf{X}$ using three distinct matrices $W_{\mathbf{Q}} \in \mathbb{R}^{d \times d'}$, $W_{\mathbf{K}} \in \mathbb{R}^{d \times d'}$ and $W_{\mathbf{V}} \in \mathbb{R}^{d \times d'}$, resulting in the representations for query ($\mathbf{Q}$), key ($\mathbf{K}$), and value ($\mathbf{V}$), which are then used to compute the output features described as follows:

$$\mathbf{Q} = \mathbf{X}W_{\mathbf{Q}}, \ \mathbf{K} = \mathbf{X}W_{\mathbf{K}}, \ \mathbf{V} = \mathbf{X}W_{\mathbf{V}},$$

$$\text{Attention}(\mathbf{X}) = \text{softmax}\left(\frac{\mathbf{Q}\mathbf{K}^\top}{\sqrt{d'}}\right)\mathbf{V}. \tag{1}$$

Technically, transformers can be considered as message-passing GNNs operating on fully-connected graphs, decoupled from the input graphs. The main research question in the context of graph transformers is how to incorporate the structural bias of the given input graphs by adapting the attention mechanism or by augmenting the original features. The **Graph Transformer (GT)** [19] represents an early work using Laplacian eigenvectors as positional encoding (PE), and various extensions and alternative models have been proposed since then [64]. For instance, the **structure-aware transformer (SAT)** [11] extracts a subgraph representation rooted at each node before computing the attention. Most initial works in the area focus on the classification of smaller graphs, such as molecules; yet, recently, **GraphGPS** [68] and follow-up works [92, 81, 80, 82, 12, 45, 72, 16, 26] also consider larger graphs.

## 2.3 Hierarchy in Graph Learning

In message passing GNNs, hierarchical pooling of node representations is a proven method for incorporating coarsening into reasoning [5, 28, 87, 48, 36, 69]. With GNNs, coarsened graph

representations are further considered in the context of molecules [42] and virtual nodes [39]. Additionally, HTAK [3] employ graph hierarchies to develop a novel graph kernel by transitively aligning the nodes across multi-level hierarchical graphs. The recent **HC-GNN** [94] demonstrates competitive performance in node classification on large-scale graphs, utilizing hierarchical community structures for message passing.

In graph transformers, there are currently only a few hierarchical models. **ANS-GT** [91] use adaptive node sampling in their graph transformer, enabling it for large-scale graphs and capturing long-range dependencies. Their model groups nodes into super-nodes and allows for interactions between them. Similarly, **HSGT** [96] aggregates multi-level graph information and employs graph hierarchical structure to construct intra-level and inter-level transformer blocks. The intra-level block facilitates the exchange and transformation of information within the local context of each node, while the inter-level block adaptively coalesces every substructure present. Our concurrent work directly incorporates hierarchy into the attention, a fundamental building block of the transformer architecture, making it flexible and applicable to existing graph transformers. Additionally, **Coarformer** [47] utilizes graph coarsening techniques to generate coarse views of the original graph, which are subsequently used as input for the transformer model. Likewise, PatchGT [27] starts by segmenting graphs into patches using spectral clustering and then learns from these non-trainable graph patches. **MGT** [66] learns atomic representations and groups them into meaningful clusters, which are then fed to a transformer encoder to calculate the graph representation. However, these approaches typically yield coarse-level representations that lack comprehensive awareness of the original node-level features [41]. In contrast, our model integrates hierarchical information from a broader distance perspective, thereby avoiding the oversimplification in these coarse-level representations.

## 3 Our Method

### 3.1 Hierarchical Distance Structural Encoding (HDSE)

Firstly, we introduce a novel concept called *graph hierarchy distance* (GHD), which is defined as follows.

**Definition 1** (**Graph Hierarchy Distance**). *Given two nodes $u, v$ in graph $G$, and a graph hierarchy $(G^i, \phi_i)_{i \geq 0}$, the $k$-level hierarchy distance between $u$ and $v$ is defined as*

$$\mathrm{GHD}^0(u, v) = \mathrm{SPD}(u, v),$$
$$\mathrm{GHD}^k(u, v) = \mathrm{SPD}(\phi_{k-1}...\phi_0(u), \phi_{k-1}...\phi_0(v)), \tag{2}$$

*where $\mathrm{SPD}(\cdot, \cdot)$ is the shortest path distance between two nodes ($\infty$ if the nodes are not connected), and $\phi_{k-1}...\phi_0(\cdot)$ maps a node in $G^0$ to a node in $G^k$.*

Note that the $k$-level hierarchy distance adheres to the definition of a metric, being zero for $v = u$, invariably positive, symmetric, and fulfilling the triangle inequality. As illustrated on the left side of Figure 1, it can be observed that $\mathrm{GHD}^0(v_1, v_{11}) = 7$, whereas $\mathrm{GHD}^1(v_1, v_{11}) = 2$.

Graph hierarchies have been observed to assist in identifying community structures in graphs that exhibit a clear property of tightly knit groups, such as social networks and citation networks [31]. They may also aid in prediction over graphs with a clear hierarchical structure, such as metal–organic frameworks or proteins. Fig. 2 shows that with the graph hierarchies generated by the Newman coarsening method, $\mathrm{GHD}^1$ is capable of capturing chemical motifs, including CF3 and aromatic rings.

Based on GHD, we propose *hierarchical distance structural encoding* (HDSE), defined for each pair of nodes $i, j \in V$:

$$\mathrm{D}_{i,j} = \left[\mathrm{GHD}^0, \mathrm{GHD}^1, ..., \mathrm{GHD}^K\right]_{i,j} \in \mathbb{R}^{K+1}, \tag{3}$$

where $\mathrm{GHD}^k$ is the $k$-level hierarchy distance matrix, and $K \in \mathbb{N}$ controls the maximum level of hierarchy considered.

We demonstrate the superior expressiveness of HDSE over SPD using recently proposed graph isomorphism tests inspired by the Weisfeiler-Leman algorithm [78]. In particular, [89] introduced the Generalized Distance Weisfeiler-Leman (GD-WL) Test and applied it to analyze a graph transformer architecture that employs $\mathrm{SPD}(i, j)$ as relative positional encodings. They proved that the graph

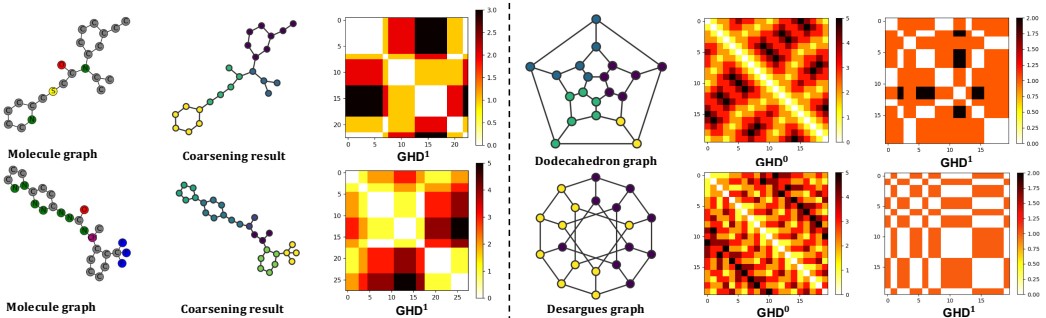

Figure 2: Examples of graph coarsening results and hierarchy distances. Left: HDSE can capture chemical motifs such as CF3 and aromatic rings on molecule graphs. Right: HDSE can distinguish the Dodecahedron and Desargues graphs. The Dodecahedral graph has 1-level hierarchy distances of length 2 (indicated by the dark color), while the Desargues graph doesn't. In contrast, the GD-WL test with SPD cannot distinguish these graphs [89].

transformer's maximum expressiveness is the GD-WL test with SPD. Here, we also use the GD-WL test to showcase the expressiveness of HDSE.

**Proposition 1** (**Expressiveness of HDSE**). *GD-WL with HDSE* $(\mathrm{D}_{i,j})$ *is strictly more expressive than GD-WL with the shortest path distance* $\mathrm{SPD}(i,j)$.

The proof is provided in Appendix 4. Firstly, we show that the GD-WL test using HDSE can differentiate between any two graphs that can be distinguished by the GD-WL test with SPD. Next, we show that the GD-WL test with HDSE is capable of distinguishing the Dodecahedron and Desargues graphs (Figure 2) while the one with SPD cannot.

### 3.2 Integrating HDSE in Graph Transformers

We integrate HDSE $(\mathrm{D}_{i,j})$ into the attention mechanism of each graph transformer layer to bias each node update. To achieve this, we use an end-to-end trainable function $\mathrm{Bias}: \mathbb{R}^{K+1} \rightarrow \mathbb{R}$ to learn the biased structure weight $\mathrm{H}_{i,j} = \mathrm{Bias}(\mathrm{D}_{i,j})$. We limit the maximum distance length to a value $L$, based on the hypothesis that detailed information loses significance beyond a certain distance. By imposing this limit, the model can extend acquired patterns to graphs of varying sizes not encountered in training. Specifically, we implement the function $\mathrm{Bias}$ using an MLP as follows:

$$\mathrm{H}_{i,j} = \mathrm{MLP}\left(\left[\mathbf{e}^0_{\mathrm{clip}^0_{i,j}}, \cdots, \mathbf{e}^K_{\mathrm{clip}^K_{i,j}}\right]\right) \in \mathbb{R},$$

$$\mathrm{clip}^k_{i,j} = \min\left(L, \mathrm{GHD}^k_{i,j}\right), 0 \leq k \leq K, \tag{4}$$

where $\left[\mathbf{e}^k_0, \mathbf{e}^k_1, \cdots, \mathbf{e}^k_L\right]_{0 \leq k \leq K} \in \mathbb{R}^{d \times (L+1)}$ collects $L+1$ learnable feature embedding vectors $\mathbf{e}^k_i$ for hierarchy level $k$. By embedding the hierarchy distances at different levels into learnable feature vectors, it may enhance the aggregation of multi-level graph information among nodes and expands the receptive field of nodes to a larger scale. We assume single-headed attention for simplified notation, but when extended to multiheaded attention, one bias is learned per distance per head.

We incorporate the learned biased structure weights $H$ to graph transformers, using the popular biased self-attention method proposed by [86], formulated as:

$$\mathrm{Attention}\left(\mathbf{X}\right) = \mathrm{softmax}\left(\mathrm{A} + \mathrm{H}\right)\mathbf{V}, \mathrm{A} = \frac{\mathbf{Q}\mathbf{K}^\top}{\sqrt{d'}}, \tag{5}$$

where the original attention score $\mathrm{A}$ is directly augmented with $\mathrm{H}$. This approach is backbone-agnostic and can be seamlessly integrated into the self-attention mechanism of existing graph transformer architectures. Notably, we have the following results on expressiveness and generalization.

**Proposition 2.** *The power of a graph transformer with HDSE to distinguish non-isomorphic graphs is at most equivalent to that of the GD-WL test with HDSE. With proper parameters and an adequate number of heads and layers, a graph transformer with HDSE can match the power of the GD-WL test with HDSE.*

See the proof in Appendix 5. This result provides a precise characterization of the expressivity power and limitations of graph transformers with HDSE. Combining Proposition 1, 2 and Proofs A.1 in [86] immediately yields the following corollary:

**Corollary 1** (**Expressiveness of Graph Transformers with HDSE**). *There exists a graph transformer using HDSE (with fixed parameters), denoted as $\mathcal{M}$, such that $\mathcal{M}$ is more expressive than graph transformers with the same architecture using SPD or using no relative positional encoding, regardless of their parameters.*

This is a fine-grained expressiveness result of graph transformers with HDSE. It demonstrate the superior expressiveness of HDSE over SPD in graph transformers.

**Proposition 3** (**Generalization of Graph Transformers with HDSE**). *(Informal) For a semi-supervised binary node classification problem, suppose the label of each node $i \in V$ is determined by node features in the "**hierarchical core neighborhood**" $S_*^i = \{j : \mathrm{D}_{i,j} = D^*\}$ for a certain $D^*$, where $\mathrm{D}_{i,j}$ is HDSE defined in (3). Then, a properly initialized one-layer graph transformer equipped with HDSE can learn such graphs with a desired generalization error, while using SPD or using no relative positional encoding cannot guarantee satisfactory generalization.*

The formal version and the proof are given in Appendix 6. Proposition 3 is a corollary and extension of Theorem 4.1 of [53] from SPD to HDSE. It indicates that learning with HDSE can capture the labeling function characterized by the hierarchical core neighborhood, which is more general and comprehensive than the core neighborhood for SPD proposed in [53].

## 3.3 Scaling HDSE to Large-scale Graphs

For large graphs with millions of nodes, training and deploying a transformer with full global attention is impractical due to the quadratic cost. Some existing graph transformers address this issue by sampling nodes for attention computation [91, 96]. While our HDSE can enhance these models, this sampling approach compromises the expressivity needed to capture interactions among all pairs of nodes. However, in the NLP domain, Linformer [77] utilizes a learnable low-rank projection matrix $\tilde{\mathbf{P}}$ to reduce the complexity of the self-attention module to linear levels:

$$\text{Attention}\left(\mathbf{X}\right) = \text{softmax}\left(\mathbf{X}\mathrm{W}_{\mathbf{Q}}(\tilde{\mathbf{P}}\mathbf{X}\mathrm{W}_{\mathbf{K}})^{\top}/\sqrt{d'}\right)\tilde{\mathbf{P}}\mathbf{X}\mathrm{W}_{\mathbf{V}}. \tag{6}$$

Inspired by Linformer, models like GOAT [45] and Gapformer [56] in the graph domain also employ projection matrices to reduce the number of nodes by projecting them onto fewer super-nodes, consequently compressing the input node feature matrix $\mathbf{X}$ into a smaller dimension. This transformation enables the aggregation of information from super-nodes and reduces the quadratic complexity of attention computation to linear complexity. Here, we can replace the projection matrix with our coarsening partition matrix $P$ (see Sec. 2.1) to obtain representations of coarser graphs at higher levels. Observe that we can still calculate meaningful distances at these higher hierarchy levels, using a *high-level* HDSE as follow:

$$\mathrm{D}_{i,j}^c = \left[\mathrm{GHD}^c(\prod_{l=0}^{c-1} P^l), ..., \mathrm{GHD}^K(\prod_{l=0}^{c-1} P^l)\right]_{i,j}  \quad 1 \le c \le K, \tag{7}$$

where each row of the projection matrix $P^l$ (see Sec. 2.1) is a one-hot vector representing the $l$-level cluster that an input node belongs to, and $\mathrm{D}^c \in \mathbb{R}^{|V^0| \times |V^c| \times (K+1-c)}$. Note that $\mathrm{GHD}^m(\prod_{l=0}^{c-1} P^l)$, $c \le m \le K$ computes distances from input nodes to clusters at $c$-level graph hierarchy (see App. A). In practice, these distances can be directly obtained by calculating the hierarchy distance between all node pairs at the $c$-level. When $c = 0$, $\mathrm{D}^c$ becomes D in Eq 3. In this way, the high-level HDSE establishes attention between nodes in the input graph $G$ and clusters at high level hierarchies. For example, we can integrate the high-level HDSE into Linformer by adapting Equation (6):

$$\text{Attention}\left(\mathbf{X}\right) = \text{softmax}\left(\frac{\mathbf{X}\mathrm{W}_{\mathbf{Q}}(\mathbf{X}^k\mathrm{W}_{\mathbf{K}})^{\top}}{\sqrt{d'}} + \mathrm{H}^k\right)\mathbf{X}^k\mathrm{W}_{\mathbf{V}},$$
$$\mathrm{H}^k = \text{Bias}(\mathrm{D}^k) \in \mathbb{R}^{|V^0| \times |V^k|}, \tag{8}$$

where $\mathbf{X}^k \in \mathbb{R}^{|V^k| \times d}$ (see Sec. 2.1) represents the features of clusters at $k$-level, and Bias : $\mathbb{R}^{K+1-k} \mapsto \mathbb{R}$ is a end-to-end trainable function as defined in Sec. 3.2.

# 4    Evaluation

We evaluate our proposed HDSE on 18 benchmark datasets, and show state-of-the-art performance in many cases. Primarily, the following questions are investigated:

- Can **HDSE improve upon existing graph transformers**, and how does the choice of **coarsening algorithm** affect performance? (**Sec. 4.2**)
- Does our **adaptation for large graphs** also **show effectiveness**, is it marked by **efficiency**, and how does **high-level HDSE** impact the performance? (**Sec. 4.3**)

## 4.1    Experiment Setting

**Datasets.** We consider various graph learning tasks from popular benchmarks as detailed below and in Appendix B.

- **Graph-level Tasks.** For graph classification and regression, we evaluate our method on five datasets from Benchmarking GNNs [20]: ZINC, MNIST, CIFAR10, PATTERN, and CLUSTER. We also test on two peptide graph benchmarks from the Long-Range Graph Benchmark [23]: Peptides-func and Peptides-struct, focusing on classifying graphs into 10 functional classes and regressing 11 structural properties, respectively. We follow all evaluation protocols suggested by [68].
- **Node Classification on Large-scale Graphs.** We consider node classification over the citation graphs Cora, CiteSeer and PubMed [44], the Actor co-occurrence graph [14], and the Squirrel and Chameleon page-page networks from [71], all of which have 1K-20K nodes. Further, we extend our evaluation to larger datasets from the Open Graph Benchmark (OGB) [35]: ogbn-arxiv, arxiv-year, ogbn-papers100M, ogbn-proteins and ogbn-products, with node numbers ranging from 0.16M to 0.1B. We maintain all the experimental settings as described in [82].

**Baselines.** We compare our method to the following prevalent GNNs: GCN [44], GIN [84], GAT [75], GatedGCN [9], GatedGCN-RWSE [22], JKNet [85], APPNP [29], PNA [15], GPRGNN [14], SIGN [70], H2GCN [95]; and other recent GNNs with SOTA performance: LINKX [55], CIN [8], GIN-AK+ [93], HC-GNN [94]. In terms of transformer models, we consider **GT**[19], Graphormer [86], SAN [46], Coarformer [47], ANS-GT [91], EGT [38], NodeFormer [81], Specformer [7], MGT [66], AGT [62], HSGT [96], Graphormer-GD [89], **SAT** [11], **GOAT** [45], Gapformer [56], Graph ViT/MLP-Mixer [34], LargeGT [21], NAGphormer [12], Exphormer [72], DRew [32], VCR-GT [26], CoBFormer [83] and recent SOTA graph transformers **GraphGPS** [68], **GRIT** [61], SGFormer [82].

**Models + HDSE.** We integrate HDSE into GT, SAT, GraphGPS, GRIT (and ANS-GT in appendix) *only modifying their self-attention module* by Eq. 5. For fair comparisons, we use the same hyperparameters (including the number of layers, hidden dimension etc.), PE and readout as the baseline transformers. Given one of the baseline transformers **M**, we denote the modified model using HDSE by **M + HDSE**. Additionally, we integrate our high-level HDSE into **GOAT**, denoted as **GOAT + HDSE**. We fix the maximum distance length $L = 30$ and vary the maximum hierarchy level $K$ in $\{0, 1, 2\}$ in all experiments. A sensitivity analysis of these two parameters is presented in Appendix C. During training, the steps of coarsening and distance calculation [17] can be treated as pre-processing, since they only need to be run once. We detail the choice and runtime of coarsening algorithms for HDSE in the appendix. Detailed experimental setup and hyperparameters are in Appendix B due to space constraints.

## 4.2    Results on Graph-level Tasks

**Benchmarks from Benchmarking GNNs, Table 2.** We observe that nearly all four baseline graph transformers, when combined with HDSE, demonstrate performance improvements. Note that the enhancement is overall especially considerable for GT. On CIFAR10, we also obtain similar improvement for GraphGPS. Among them, GT shows the greatest enhancement and becomes competitive to more complex models. Our model attains the best or second-best mean performance for all datasets. While the improvement for GRIT is smaller, as its relative random walk probabilities (RRWP) already incorporate distance information [61], we still observe improvements in three datasets. This indicates that HDSE can provide additional information beyond what is captured by RRWP. Notably, it is observed that the SOTA SGFormer tailored for large-scale node classification underperforms in graph-level tasks.

Table 2: Test performance in five benchmarks from [20]. The results are presented as the mean ± standard deviation from 5 runs using different random seeds. Baseline results were obtained from their respective original papers. * indicates a statistically significant difference against the baseline w/o HDSE from the one-tailed t-test. Highlighted are the top **first**, **second** and **third** results.

| Model | ZINC MAE ↓ | MNIST Accuracy ↑ | CIFAR10 Accuracy ↑ | PATTERN Accuracy ↑ | CLUSTER Accuracy ↑ |
|---|---|---|---|---|---|
| GCN | 0.367 ± 0.011 | 90.705 ± 0.218 | 55.710 ± 0.381 | 71.892 ± 0.334 | 68.498 ± 0.976 |
| GIN | 0.526 ± 0.051 | 96.485 ± 0.252 | 55.255 ± 1.527 | 85.387 ± 0.136 | 64.716 ± 1.553 |
| GatedGCN | 0.282 ± 0.015 | 97.340 ± 0.143 | 67.312 ± 0.311 | 85.568 ± 0.088 | 73.840 ± 0.326 |
| PNA | 0.188 ± 0.004 | 97.940 ± 0.120 | 70.350 ± 0.630 | – | – |
| CIN | 0.079 ± 0.006 | – | – | – | – |
| GIN-AK+ | 0.080 ± 0.001 | – | 72.190 ± 0.130 | 86.850 ± 0.057 | – |
| SGFormer | 0.306 ± 0.023 | – | – | 85.287 ± 0.097 | 69.972 ± 0.634 |
| SAN | 0.139 ± 0.006 | – | – | 86.581 ± 0.037 | 76.691 ± 0.650 |
| Graphormer-GD | 0.081 ± 0.009 | – | – | – | – |
| Specformer | 0.066 ± 0.003 | – | – | – | – |
| EGT | 0.108 ± 0.009 | 98.173 ± 0.087 | 68.702 ± 0.409 | 86.821 ± 0.020 | 79.232 ± 0.348 |
| Graph ViT/MLP-Mixer | 0.073 ± 0.001 | 97.422 ± 0.110 | 73.961 ± 0.330 | – | – |
| Exphormer | - | 98.550 ± 0.039 | 74.696 ± 0.125 | 86.742 ± 0.015 | 78.071 ± 0.037 |
| GT | 0.226 ± 0.014 | 90.831 ± 0.161 | 59.753 ± 0.293 | 84.808 ± 0.068 | 73.169 ± 0.622 |
| **GT + HDSE** | 0.159 ± 0.006* | 94.394 ± 0.177* | 64.651 ± 0.591* | 86.713 ± 0.049* | 74.223 ± 0.573* |
| SAT | 0.094 ± 0.008 | – | – | 86.848 ± 0.037 | 77.856 ± 0.104 |
| **SAT + HDSE** | 0.084 ± 0.003* | – | – | 86.933 ± 0.039* | 78.513 ± 0.097* |
| GraphGPS | 0.070 ± 0.004 | 98.051 ± 0.126 | 72.298 ± 0.356 | 86.685 ± 0.059 | 78.016 ± 0.180 |
| **GraphGPS + HDSE** | 0.062 ± 0.003* | 98.367 ± 0.106* | 76.180 ± 0.277* | 86.737 ± 0.055 | 78.498 ± 0.121* |
| GRIT | 0.059 ± 0.002 | 98.108 ± 0.111 | 76.468 ± 0.881 | 87.196 ± 0.076 | 80.026 ± 0.277 |
| **GRIT + HDSE** | 0.059 ± 0.004 | 98.424 ± 0.124* | 76.473 ± 0.429 | 87.281 ± 0.064 | 79.965 ± 0.203 |

Table 3: Test performance on two peptide datasets from Long-Range Graph Benchmarks (LRGB) [23].

| Model | Peptides-func AP ↑ | Peptides-struct MAE ↓ |
|---|---|---|
| GCN | 0.5930 ± 0.0023 | 0.3496 ± 0.0013 |
| GINE | 0.5498 ± 0.0079 | 0.3547 ± 0.0045 |
| GatedGCN | 0.5864 ± 0.0035 | 0.3420 ± 0.0013 |
| GatedGCN+RWSE | 0.6069 ± 0.0035 | 0.3357 ± 0.0006 |
| GT | 0.6326 ± 0.0126 | 0.2529 ± 0.0016 |
| SAN+RWSE | 0.6439 ± 0.0075 | 0.2545 ± 0.0012 |
| MGT+WavePE | 0.6817 ± 0.0064 | 0.2453 ± 0.0025 |
| GRIT | 0.6988 ± 0.0082 | 0.2460 ± 0.0012 |
| Exphormer | 0.6527 ± 0.0043 | 0.2481 ± 0.0007 |
| Graph ViT/MLP-Mixer | 0.6970 ± 0.0080 | 0.2475 ± 0.0015 |
| DRew | 0.7150 ± 0.0044 | 0.2536 ± 0.0015 |
| GraphGPS | 0.6535 ± 0.0041 | 0.2500 ± 0.0012 |
| **GraphGPS + HDSE** | 0.7156 ± 0.0058* | 0.2457 ± 0.0013* |

Table 4: Ablation experiments of coarsening algorithms on ZINC.

| Model | Coarsening algorithm | ZINC MAE ↓ |
|---|---|---|
| | w/o | 0.094 ± 0.008 |
| SAT | METIS | 0.089 ± 0.005 |
| | Spectral | 0.088 ± 0.004 |
| | Loukas | 0.084 ± 0.003 |
| | Newman | 0.087 ± 0.002 |
| | Louvain | 0.088 ± 0.003 |
| | w/o | 0.070 ± 0.004 |
| GraphGPS | METIS | 0.069 ± 0.002 |
| | Spectral | 0.063 ± 0.003 |
| | Loukas | 0.067 ± 0.002 |
| | Newman | 0.062 ± 0.003 |
| | Louvain | 0.064 ± 0.002 |

**Long-Range Graph Benchmark, Table 3.** We consider GraphGPS due to its superior performance. Note that our HDSE module only introduces a small number of additional parameters, allowing it to remain within the benchmark's ∼500k model parameter budget. In the Peptides-func dataset, HDSE yields a significant improvement of 6.21%. This is a promising result and hints at potentially great benefits for macromolecular data more generally.

**Ablation Study and Visualization, Table 4, 13, 16, Figure 3, 5.** First, we conduct several ablation experiments of coarsening algorithms on ZINC and observe that the dependency on the coarsening varies with the transformer backbone. For instance, the multi-level graph structures extracted by the Newman algorithm yields the largest improvement for GraphGPS. More generally, our experiments indicate that Newman works best for molecular graphs. We visualize the attention scores on the ZINC and Peptides-func datasets respectively, as shown in Figure 3. The results indicate that our HDSE method successfully leverages hierarchical structure.

We also conduct a sensitivity analysis on maximal hierarchy level $K$ and maximum distance length $L$ in Appendix C. The variation in the optimal $K$ and $L$ could stem from the distinct hierarchical structures inherent in different graphs.

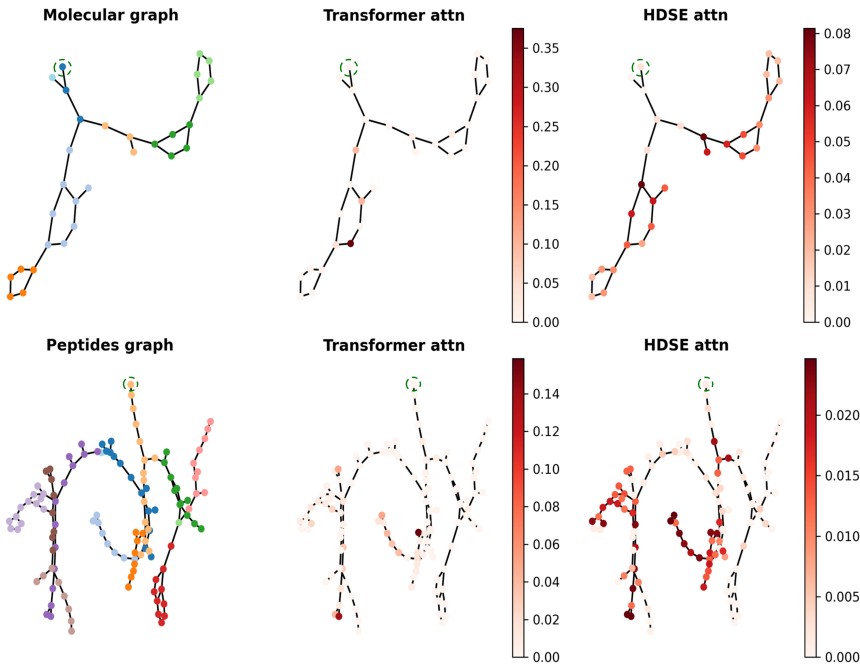

Figure 3: Visualization of attention weights for the transformer attention and HDSE attention. The left side illustrates the graph coarsening result. The center column displays the attention weights of a sample node learned by the classic GT [19], while the right column showcases the attention weights learned by the HDSE attention.

Table 5: **Node classification** on large-scale graphs (%). The baseline results were primarily taken from [82], with the remaining obtained from their respective original papers. OOM indicates out-of-memory when training on a GPU with 24GB memory.

| Model | Cora | CiteSeer | PubMed | Actor | Squirrel | Chameleon | ogbn-proteins | ogbn-arxiv | arxiv-year | ogbn-products | ogbn-100M |
|---|---|---|---|---|---|---|---|---|---|---|---|
| # nodes | 2,708 | 3,327 | 19,717 | 7,600 | 2223 | 890 | 132,534 | 169,343 | 169,343 | 2,449,029 | 111,059,956 |
| # edges | 5,278 | 4,552 | 44,324 | 29,926 | 46,998 | 8,854 | 39,561,252 | 1,166,243 | 1,166,243 | 61,859,140 | 1,615,685,872 |
| | Accuracy↑ | Accuracy↑ | Accuracy↑ | Accuracy↑ | Accuracy↑ | Accuracy↑ | ROC-AUC↑ | Accuracy↑ | Accuracy↑ | Accuracy↑ | Accuracy↑ |
| SIGN | 82.1 ± 0.3 | 72.4 ± 0.8 | 79.5 ± 0.5 | 36.5 ± 1.0 | 40.7 ± 2.5 | 41.7 ± 2.2 | 71.24 ± 0.46 | 71.95 ± 0.11 | - | 80.52 ± 0.16 | **65.11 ± 0.14** |
| LINKX | - | - | - | 36.1 ± 1.5 | **41.9 ± 1.2** | **43.8 ± 2.9** | 66.18 ± 0.33 | 53.53 ± 0.36 | 71.59 ± 0.71 | - | - |
| HC-GNN | 81.9 ± 0.4 | **72.5 ± 0.6** | **80.2 ± 0.6** | - | - | - | - | **72.79 ± 0.25** | - | - | - |
| Graphormer | 75.8 ± 1.1 | 65.6 ± 0.6 | OOM | OOM | 40.9 ± 2.5 | 41.9 ± 2.8 | OOM | OOM | OOM | OOM | OOM |
| SAT | 72.4 ± 0.3 | 60.9 ± 1.3 | OOM | OOM | - | - | OOM | OOM | OOM | OOM | OOM |
| ANS-GT | 79.4 ± 0.9 | 64.5 ± 0.7 | 77.8 ± 0.7 | 35.2 ± 1.3 | 40.8 ± 2.1 | 42.6 ± 2.7 | 74.67 ± 0.65 | 72.34 ± 0.50 | - | 80.64 ± 0.29 | - |
| AGT | 81.7 ± 0.4 | 71.0 ± 0.6 | - | - | - | - | - | 72.28 ± 0.38 | 47.38 ± 0.78 | - | - |
| HSGT | **83.6 ± 1.8** | 67.4 ± 0.9 | 79.7 ± 0.5 | - | - | - | 78.13 ± 0.25 | 72.58 ± 0.31 | - | **81.15 ± 0.13** | - |
| GraphGPS | 76.5 ± 0.6 | - | 65.7 ± 1.0 | 33.1 ± 0.8 | - | 36.2 ± 0.6 | - | 70.97 ± 0.41 | - | OOM | OOM |
| Gapformer | 83.5 ± 0.4 | 71.4 ± 0.6 | **80.2 ± 0.4** | - | - | - | - | 71.90 ± 0.19 | - | - | - |
| LargeGT | - | - | - | - | - | - | - | - | - | - | 64.73 ± 0.05 |
| VCR-GT | - | - | - | - | - | - | - | - | **54.15 ± 0.09** | - | - |
| NAGphormer | - | - | - | 34.3 ± 0.9 | 39.7 ± 0.8 | 40.3 ± 1.7 | - | 70.13 ± 0.55 | - | 73.55 ± 0.21 | - |
| Exphormer | - | - | - | - | - | - | - | 72.44 ± 0.28 | - | OOM | OOM |
| NodeFormer | 82.2 ± 0.9 | **72.5 ± 1.1** | 79.9 ± 1.0 | 36.9 ± 1.0 | 38.5 ± 1.5 | 34.7 ± 4.1 | 77.45 ± 1.15 | - | - | 72.93 ± 0.13 | - |
| CoBFormer | - | - | - | **37.4 ± 1.0** | - | - | - | **73.17 ± 0.18** | - | 78.15 ± 0.07 | - |
| SGFormer | **84.5 ± 0.8** | 72.6 ± 0.2 | 80.3 ± 0.6 | 37.9 ± 1.1 | 41.8 ± 2.2 | 44.9 ± 3.9 | 79.53 ± 0.38 | 72.63 ± 0.13 | - | 75.36 ± 0.33 | **66.01 ± 0.37** |
| GOAT | 82.1 ± 0.9 | 71.6 ± 1.3 | 78.9 ± 1.5 | 32.1 ± 1.8 | 41.1 ± 2.5 | 43.3 ± 4.3 | **78.37 ± 0.26** | 72.41 ± 0.40 | 53.57 ± 0.18 | **82.00 ± 0.43** | 65.05 ± 0.13 |
| **GOAT + HDSE** | **83.9 ± 0.7*** | 73.1 ± 0.7 | 80.6 ± 1.0 | **38.0 ± 1.5*** | 43.2 ± 2.4 | 46.0 ± 3.2 | **80.34 ± 0.32*** | **73.26 ± 0.19*** | **54.23 ± 0.26*** | **83.38 ± 0.17*** | **66.04 ± 0.15*** |

**Synthetic Community Graphs, Table 19.** We evaluate our methods on the community datasets from [88], generated using the Erdos-Renyi model [24]. These graphs have adjacency matrices that obey the certain clustered structure. As evidenced in Table 19, the GT struggles to detect such structures; and solely utilizing SPD proves inadequate; however, our HDSE, enriched with coarsening structural information, effectively captures these structures.

## 4.3 Results on Large-scale Graphs

**Overall Performance, Table 5.** We select four categories of baselines: GNNs, graph transformers with proven performance on graph-level tasks, graph transformers with hierarchy, and scalable graph transformers. It is noteworthy that while some graph transformers exhibit superior performance on graph-level tasks, they consistently result in out-of-memory (OOM) in large-scale node tasks. The results are remarkably consistent. In relatively smaller datasets graphs (on the left side), the integra-

Table 6: Efficiency comparison of GOAT + HDSE and scalable graph transformer competitors; training time per epoch.

|  | PubMed | ogbn-proteins | ogbn-arxiv | ogbn-products | ogbn-papers100M |
|---|---|---|---|---|---|
| NodeFormer | 321.4ms | 1.8s | 0.6s | 5.6s | 595.1s |
| SGFormer | 15.4ms | 0.8s | 0.2s | 4.8s | 579.4s |
| GOAT+HDSE | 13.2ms | 0.6s | 0.2s | 5.3s | 446.5s |

Table 7: Ablation study of GOAT + HDSE. "w/o coarsening" refers to replacing the projection matrix with the original projection matrix used in GOAT.

|  | Actor ↑ | ogbn-proteins ↑ | arxiv-year ↑ |
|---|---|---|---|
| GOAT+HDSE | 38.0 ± 1.5 | 80.3 ± 0.3 | 54.2 ± 0.2 |
| w/o HDSE | 34.6 ± 2.2 | 79.4 ± 0.3 | 53.6 ± 0.3 |
| w/o coarsening | 32.1 ± 1.8 | 78.3 ± 0.4 | 53.5 ± 0.2 |

tion of high-level HDSE enables GOAT to demonstrate competitive performance among baseline models. This could be due to the coarsening projection filtering out the edges from neighboring nodes of different categories and providing a global perspective enriched with multi-level structural information. For all larger graphs (on the right side), our high-level HDSE method significantly enhances GOAT's performance beyond its original version. This indicates that the structural bias provided by graph hierarchies is capable of handling the node classification task in such larger graphs and effectively retains global information. We investigated this in more detail in our ablation experiments. Furthermore, we also observed that all graph transformers with hierarchy suffer from serious overfitting, attributed to their relatively complex architectures. In contrast, our model's simple architecture contributes to its better generalization.

**Efficiency Comparison, Table 6.** We report the efficiency results on PubMed, ogbn-proteins, ogbn-arxiv, ogbn-products and ogbn-100M. It is easy to see that our model outperforms NodeFormer in speed, matching the pace of the latest and fastest model, SGFormer [82]. It achieves true linear complexity with a streamlined architecture.

**Ablation Study, Table 7.** To determine the utility of our architectural design choices, we conduct ablation experiments on GOAT + HDSE over three datasets. The results presented in Table 7, include (1) removing the high-level HDSE and (2) replacing the coarsening projection matrix with the original projection matrix used in GOAT. These experiments reveal a decline in all performance, thereby validating the effectiveness of our architectural design.

## 5 Conclusions

We have introduced the Hierarchical Distance Structural Encoding (HDSE) method to enhance the capabilities of transformer architectures in graph learning tasks. We have developed a flexible framework to integrate HDSE with various graph transformers. Further, for applying graph transformers with HDSE to large-scale graphs, we have introduced a high-level HDSE approach that effectively biases linear transformers towards the multi-level structure. Theoretical analysis and empirical results validate the effectiveness and generalization capabilities of HDSE, demonstrating its potential for various real-world applications.

## Acknowledgments and Disclosure of Funding

We extend our sincere gratitude to Veronika Thost, Yicheng Pan, Zixu Zhao and Jaan Li for their invaluable guidance in our experiments. We also express our appreciation to all the anonymous reviewers and ACs for their insightful and constructive feedback. This work received support from National Key R&D Program of China (2021YFB3500700), NSFC Grant 62172026, National Social Science Fund of China 22&ZD153, the Fundamental Research Funds for the Central Universities, State Key Laboratory of Complex & Critical Software Environment (CCSE), HK PolyU Grant P0051029, HK PolyU Grant P0038850, and HK ITF Grant ITS/359/21FP. Lei Shi is with Beihang University and State Key Laboratory of Complex & Critical Software Environment.

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

# A   Proof

**Proposition 4.** *(Restatement of Proposition 1) GD-WL with HDSE* $(D_{i,j})$ *is strictly more expressive than GD-WL with shortest path distances* $(\text{SPD}(i,j))$.

*Proof.* First, we show that GD-WL with HDSE is at least as expressive as GD-WL with shortest path distances (SPD). Then, we provide a specific example of two graphs that cannot be distinguished by GD-WL with SPD, but can be distinguished by GD-WL with HDSE.

Let $\text{SPD}(i,j) \in \mathbb{R}$ denote the encoding for shortest path distance. It is worth mentioning that

$$D_{i,j,0} = \text{GHD}^0(i,j) = \text{SPD}(i,j).$$

Thus, $D_{i,j}$ is a function of $\text{SPD}(i,j)$, and hence $D_{i,j}$ refines $\text{SPD}(i,j)$. To conclude this, we utilize Lemma 2 from [4], which states that refinement is maintained when using multisets of colors. This observation confirms that GD-WL with HDSE is at least as powerful as GD-WL with SPD.

To show that GD-WL with HDSE is strictly more expressive, we provide an example of two non-isomorphic graphs that can be distinguished by the HDSE but not the SPD: the Desargues graph and the Dodecahedral graph. As depicted in Figure 6 of [89], it has been observed that GD-WL with SPD fails to distinguish these graphs. However, GD-WL with our HDSE can. Figure 4 shows the coarsening results of the Girvan-Newman Algorithm [31]. We can demonstrate that, for the Dodecahedral graph, each node has 1-level hierarchy distances of length 2 to other nodes. On the other hand, in the Desargues graph, there are no such distances of length 2 between any pair of nodes. $\square$

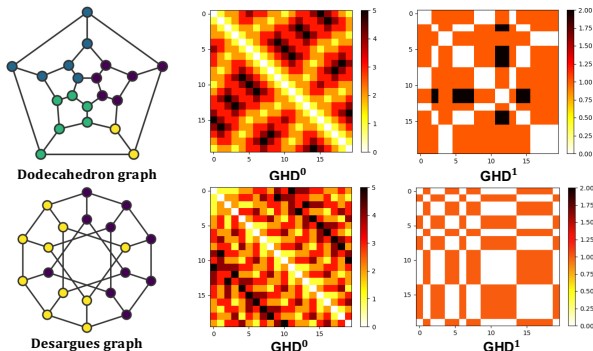

Figure 4: GD-WL with HDSE can distinguish Dodecahedron and Desargues graphs, but GD-WL with SPD cannot.

**Proposition 5.** *(Restatement of Corollary 2) The power of a graph transformer with HDSE to distinguish non-isomorphic graphs is at most equivalent to that of the GD-WL test with HDSE. With proper parameters and an adequate number of heads and layers, a graph transformer with HDSE can match the power of the GD-WL test with HDSE.*

*Proof.* The theorem is divided into two parts: the first and second halves. We begin by considering the first half: The power of a graph transformer with HDSE to distinguish non-isomorphic graphs is at most equivalent to that of the GD-WL test with HDSE.

Recall that the GD-WL with HDSE is quite straightforward and can be expressed as:

$$\chi_G^t(v) := \text{hash}\left\{(D_{v,u}, \chi_G^{t-1}(u)) : u \in V\right\}$$

where $\chi_G^t(v)$ represents a color mapping function.

Suppose after $t$ iterations, a graph transformer with HDSE $\mathcal{M}$ has $\mathcal{M}(G_1) \neq \mathcal{M}(G_2)$, yet GD-WL with HDSE fails to distinguish $G_1$ and $G_2$ as non-isomorphic. It implies that from iteration 0 to $t$ in the GD-WL test, $G_1$ and $G_2$ always have the same collection of node labels. Particularly, since $G_1$ and $G_2$ have the same GD-WL node labels for iterations $i + 1$ for any $i = 0, \ldots, t - 1$, they also

share the same collection of GD-WL node labels $\left\{(\mathrm{D}_{v,u}, \chi_G^i(u)) : u \in V\right\}$. Otherwise, the GD-WL test would have produced different node labels at iteration $i + 1$ for $G_1$ and $G_2$.

We show that within the same graph, for example $G = G_1$, if GD-WL node labels $\chi_G^i(v) = \chi_G^i(w)$, then the graph transformer node features $h_v^i = h_w^i$ for any iteration $i$. This is clearly true for $i = 0$ because GD-WL and graph transformer start with identical node features. Assuming this holds true for iteration $j$, if for any $v, w$, $\chi_G^{j+1}(v) = \chi_G^{j+1}(w)$, then we must have

$$\left\{(\mathrm{D}_{v,u}, \chi_G^j(u)) : u \in V\right\} = \left\{(\mathrm{D}_{w,u}, \chi_G^j(u)) : u \in V\right\}.$$

By our assumption at iteration $j$, we deduce that

$$h_v^{j+1} = \sum_{u \in V} \mathrm{softmax}(\mathrm{Bias}(\mathrm{D}_{v,u}) + h_v^j \mathbf{W_Q} (h_u^j \mathbf{W_K})^\top) h_u^j \mathbf{W_V} = \phi(\left\{(\mathrm{D}_{v,u}, \chi_G^j(u)) : u \in V\right\}).$$

Hence,

$$h_v^{j+1} = \phi(\left\{(\mathrm{D}_{v,u}, \chi_G^j(u)) : u \in V\right\}) = \phi(\left\{(\mathrm{D}_{w,u}, \chi_G^j(u)) : u \in V\right\}) = h_w^{j+1}.$$

By induction, if GD-WL node labels $\chi_G^i(v) = \chi_G^i(w)$, we always have the graph transformer node features $h_v^i = h_w^i$ for any iteration $i$. Consequently, from $G_1$ and $G_2$ having identical GD-WL node labels, it follows that they also have the same graph transformer node features.

Therefore, $h_v^{i+1} = h_w^{i+1}$. Given that the graph-level readout function is permutation-invariant with respect to the collection of node features, $\mathcal{M}(G_1) = \mathcal{M}(G_2)$. This leads to a contradiction.

This completes the proof of the first half of the theorem. For the theorem's second half, we can entirely leverage the proof of Theorem E.3 by [89] (provided in Appendix E.3), which presents a similar situation.

$\square$

**Proposition 6.** *(Full version of Proposition 3) For a semi-supervised binary node classification problem, suppose the label of each node $i \in V$ in the whole graph is determined by the majority vote of discriminative node features in the "hierarchical core neighborhood": $S_*^i = \{j : \mathrm{D}_{i,j} = D^*\}$ for a certain $D^*$, where $\mathrm{D}_{i,j}$ is HDSE defined in (3). Assume noiseless node features. Then, as long as the model is large enough, the batch size $B \geq \Omega(\epsilon^{-2})$, the step size $\eta < 1$, the number of iterations $T$ satisfies $T = \Theta(\eta^{-1/2})$ and the number of known labels satisfies $|\mathcal{L}| \geq \max\{\Omega((1 + \delta_{D^*}^2) \cdot \log N), BT\}$, where $\delta_{D^*}$ measures the maximum number of nodes in the hierarchical core neighborhood $S_*^n$ for all nodes $n$, then with a probability of at least 0.99, the returned one-layer graph transformer with HDSE trained by the SGD variant Algorithm 1 and Hinge loss in [53] can achieve a generalization error which is at most $\epsilon$ for any $\epsilon > 0$. However, we do not have a generalization guarantee to learn such a graph characterized by the hierarchical core neighborhood with a one-layer graph transformer with SPD encoding or without any relative positional encoding.*

Before starting the proof, we first briefly introduce and extend some notions and setups used in [53]. **The major differences are that (1) we extend their core neighborhood from based on SPD to HDSE (2) we use HDSE in the transformer by encoding it as a one-hot encoding for simplicity of analysis.**

Their work focuses on a semi-supervised binary node classification problem on structured graph data, where each node feature corresponds to either a discriminative or a non-discriminative feature, and the dominant discriminative feature in the core neighborhood determines each ground truth node label. The node features are sampled from a set of orthonormal vectors following other feature-learning works [40, 50, 2, 37, 52, 51, 90, 13] in theoretically analyzing Transformers. For each node, the neighboring nodes with features consistent with the label are called class-relevant nodes, while nodes with features opposite to the label are called confusion nodes. Denote the class-relevant and confusion nodes set for node $n$ as $\mathcal{D}_*^n$ and $\mathcal{D}_\#^n$, respectively. A new definition here is the distance-$D$ neighborhood $\mathcal{N}_D^n$, which is the set of nodes $\{j : \mathrm{D}_{n,j} = D, j \in V\}$. D is the HDSE defined in (3). Then, by following Definition 1 in [53], we define the winning margin for each node $n$ of distance $D$ as $\Delta_n(D) = |\mathcal{D}_*^n \cap \mathcal{N}_D^n| - |\mathcal{D}_\#^n \cap \mathcal{N}_D^n|$. The core distance $D^*$ is the distance $D$ where the average winning margin over all nodes is the largest. We call the set of neighboring nodes

$S_*^n = \{j : D_{n,j} = D^*\}$ the core neighborhood. We then make the assumption that $\Delta_n(D^*) > 0$ for all nodes $n \in V$, following Assumption 1 in [53]. The one-layer transformer we study is formulated as

$$F(h_n) = \text{Relu}(\sum_{u \in V} \text{softmax}(\text{B}(D_{n,u})^\top b + h_n \mathbf{W_Q}(h_u \mathbf{W_K})^\top)h_u \mathbf{W_V}\mathbf{W_O})a \quad (9)$$

where $\boldsymbol{W}_O \in \mathbb{R}^{d' \times d''}$ and $\boldsymbol{a} \in \mathbb{R}^{d''}$ are the hidden and output weights in the two-layer feedforward network, and $b \in \mathbb{R}^Z$ is the trainable parameter to learn the relative positional encoding. The one-hot relative positional encoding $\text{B}(D_{n,u})$ is defined as

$$\text{B}(D_{n,u}) = \boldsymbol{c}_s, \quad (10)$$

where $\boldsymbol{c}_s$ is the $s$-th standard basis in $\mathbb{R}^Z$. $Z$ is the total number of all possible values of $D_{n,u}$ for $n, u \in V$. $\text{B}(\cdot)$ is a bijection from $\{d \in \mathbb{R}^{K+1} : d = D_{n,u}, \text{for certain } n, v \in V\}$ to $\{\boldsymbol{c}_1, \boldsymbol{c}_2, \cdots, \boldsymbol{c}_Z\}$.

Then, we provide the proof for Proposition 6.

*Proof.* The proof follows Theorem 4.1 in [53] given the above reformulation. Note that (10) can also map the SPD relationship, which is a special one-dimensional case of $D_{v,u}$, between nodes as (2) in [53] by the definition of itself. It means that (9) with HDSE can achieve a generalization performance on the graph characterized by the core neighborhood as good as in [53].

However, we cannot have an inverse conclusion, i.e., providing a generalization guarantee on the graph characterized by the hierarchical core neighborhood using (9) with SPD. This is because SPD cannot distinguish nodes with the same SPD but different HDSE to a certain node. Hence, for a certain node $i \in V$, aggregating nodes using SPD may include nodes outside the hierarchical core neighborhood, of which the labels are inconsistent with the node $i$, and lead to a wrong prediction. Likewise, we cannot guarantee the generalization using a Graph Transformer without any relative positional encoding since such a model cannot distinguish nodes with different HDSE. $\square$

**Proposition 7.** *For a semi-supervised binary node classification problem, suppose the label of each node $i \in V$ in the whole graph is determined by the majority vote of discriminative node features in the "core neighborhood": $S_*^i = \{j : D_{i,j} = D^*\}$ for a certain $D^*$, where $D_{i,j}$ is HDSE defined in (3). Then, for each node $i \in V$, a properly initialized one-layer graph transformer (i) **without HDSE** (ii) and **only aggregate nodes from** $S_*^i$ can achieve the same generalization error as learning with a one-layer graph transformer (a) **with HDSE** (b) **aggregate all nodes in the graph without knowing** $S_i^*$ **in prior**.*

*Proof.* The proof follows Theorem 4.3 in [53]. When $\boldsymbol{b} = 0$ is fixed during the training, but the nodes used for training and testing in aggregation for node $n$ are subsets of $\mathcal{N}_{D^*}^n$, the bound for attention weights on class-relevant nodes is still the same as in (63) and (64) of [53]. Given a known core neighborhood $S_*^n$, the remaining parameters follow the same order-wise update as Lemmas 4, 5, and 7. The remaining proof steps follow the contents in the proof of Theorem 4.1 of [53], which leads to a generalization on a one-layer transformer with HDSE and aggregation with all nodes in the graph.

$\square$

**Explanation of** GHD **Computation in Equation 7.** As defined in Eq. 2, $\text{GHD}^m \in \mathbb{R}^{|V| \times |V|}$ represents the shortest path distance between any two input nodes at the $m$-level graph hierarchy. $\forall m, \text{GHD}^m$ has the same size as $\text{GHD}^0$.

In Eq. 7, our high-level HDSE computes, at each level $c \leq m \leq K$, distances between input nodes and clusters obtained by coarsening (i.e., super nodes at the $c$-level graph hierarchy). This is achieved by multiplying the projection matrices $\prod_{l=0}^{c-1} P^l$ to $\text{GHD}^m$. In effect, it is equivalent to selecting corresponding columns from $\text{GHD}^m$. For instance, referring to Figure 1, $\text{GHD}^1 P^0 \in \mathbb{R}^{11 \times 3}$ calculates the distances from input nodes to the super nodes at 1-level graph hierarchy, essentially selecting the first, fourth, and tenth columns from $\text{GHD}^1$.

Likewise, $\text{GHD}^m(\prod_{l=0}^{c-1} P^l) \in \mathbb{R}^{|V| \times |V^c|}$ selects $|V^c|$ columns from $\text{GHD}^m$ to represent the distances, at the $m$-level graph hierarchy, between the input nodes and the $c$-level super nodes (i.e., clusters obtained through coarsening).

# B  Experimental Details

## B.1  Computing Environment

Our implementation is based on PyG [25] and DGL [76]. The experiments are conducted on a single workstation with 4 RTX 3090 GPUs and a quad-core CPU.

Table 8: Overview of the graph learning dataset used in this work [20, 23, 44, 14, 67, 71, 35, 63, 49].

| Dataset | # Graphs | Avg. # nodes | Avg. # edges | # Feats | Prediction level | Prediction task | Metric |
|---|---|---|---|---|---|---|---|
| ZINC | 12,000 | 23.2 | 24.9 | 28 | graph | regression | MAE |
| MNIST | 70,000 | 70.6 | 564.5 | 3 | graph | 10-class classif. | Accuracy |
| CIFAR10 | 60,000 | 117.6 | 941.1 | 5 | graph | 10-class classif. | Accuracy |
| PATTERN | 14,000 | 118.9 | 3,039.3 | 3 | node | binary classif. | Accuracy |
| CLUSTER | 12,000 | 117.2 | 2,150.9 | 7 | node | 6-class classif. | Accuracy |
| Peptides-func | 15,535 | 150.9 | 307.3 | 9 | graph | 10-task classif. | AP |
| Peptides-struct | 15,535 | 150.9 | 307.3 | 9 | graph | 11-task regression | MAE |
| Cora | 1 | 2,708 | 5,278 | 2,708 | node | 7-class classif. | Accuracy |
| Citeseer | 1 | 3,327 | 4,522 | 3,703 | node | 6-class classif. | Accuracy |
| Pubmed | 1 | 19,717 | 44,324 | 500 | node | 3-class classif. | Accuracy |
| Actor | 1 | 7,600 | 26,659 | 931 | node | 5-class classif. | Accuracy |
| Squirrel | 1 | 5,201 | 216,933 | 2,089 | node | 5-class classif. | Accuracy |
| Chameleon | 1 | 2,277 | 36,101 | 2,325 | node | 5-class classif. | Accuracy |
| ogbn-proteins | 1 | 132,534 | 39,561,252 | 8 | node | 112 binary classif. | ROC-AUC |
| ogbn-arxiv | 1 | 169,343 | 1,166,243 | 128 | node | 40-class classif. | Accuracy |
| arxiv-year | 1 | 169,343 | 1,166,243 | 128 | node | 5-class classif. | Accuracy |
| ogbn-products | 2 | 2,449,029 | 61,859,140 | 100 | node | 47-class classif. | Accuracy |
| ogbn-papers100M | 1 | 111,059,956 | 1,615,685,872 | 128 | node | 172-class classif. | Accuracy |

## B.2  Description of Datasets

Table 8 presents a summary of the statistics and characteristics of the datasets. The initial five datasets are sourced from [20], followed by two from [23], and finally the remaining datasets are obtained from [44, 14, 67, 71, 35, 63, 49].

- ZINC, MNIST, CIFAR10, PATTERN, CLUSTER, Peptides-func and Peptides-struct. For each dataset, we follow the standard train/validation/test splits and evaluation metrics in [68]. For more comprehensive details, readers are encouraged to refer to [68].

- Cora, Citeseer, Pubmed, Actor, Squirrel, Chameleon, ogbn-proteins, ogbn-arxiv, ogbn-products and ogbn-papers100M. For each dataset, we use the same train/validation/test splits and evaluation metrics as [82]. For detailed information on these datasets, please refer to [82].

- Arxiv-year is a citation network among all computer science arxiv papers, as described by [55]. In this network, each node corresponds to an arxiv paper, and the edges indicate the citations between papers. Each paper is associated with a 128-dimensional feature vector, obtained by averaging the word embeddings of its title and abstract. The word embeddings are generated using the WORD2VEC model. The labels of arxiv-year are publication years clustered into fve intervals. We use the public splits shared by [55], with a train/validation/test split ratio of 50%/25%/25%.

## B.3  Hyperparameter and Reproducibility

**Models + HDSE.** For fair comparisons, we use the same hyperparameters (including training schemes, optimizer, number of layers, hidden dimension etc.) as baseline models for all of our HDSE versions. Taking GraphGPS + HDSE as an example, Tables 9 and 10 showcase the corresponding hyperparameters and coarsening algorithms. It is important to note that our HDSE module introduces only a small number of additional parameters. And each experiment is repeated 5 times to get the mean value and error bar.

**GOAT + HDSE.** To accelerate training, we do not adopt the neighbor sampling (NS) method from GOAT to sample neighbors; instead, we train directly on the entire graph. For graphs with over one million nodes, we randomly sample nodes within the graph and select their induced subgraph for batch training. For the hyperparameter selections of our high-level HDSE model, in addition to what

Table 9: Hyperparameters of GraphGPS + HDSE for five datasets from [20].

| Hyperparameter | ZINC | MNIST | CIFAR10 | PATTERN | CLUSTER |
|---|---|---|---|---|---|
| # GPS Layers | 10 | 3 | 3 | 6 | 16 |
| Hidden dim | 64 | 52 | 52 | 64 | 48 |
| GPS-MPNN | GINE | GatedGCN | GatedGCN | GatedGCN | GatedGCN |
| GPS-GlobAttn | Transformer | Transformer | Transformer | Transformer | Transformer |
| # Heads | 4 | 4 | 4 | 4 | 8 |
| Attention dropout | 0.5 | 0.5 | 0.5 | 0.5 | 0.5 |
| Graph pooling | sum | mean | mean | – | – |
| Positional Encoding | RWSE-20 | LapPE-8 | LapPE-8 | LapPE-16 | LapPE-10 |
| PE dim | 28 | 8 | 8 | 16 | 16 |
| PE encoder | linear | DeepSet | DeepSet | DeepSet | DeepSet |
| Batch size | 32 | 16 | 16 | 32 | 16 |
| Learning Rate | 0.001 | 0.001 | 0.001 | 0.0005 | 0.0005 |
| # Epochs | 2000 | 100 | 200 | 100 | 100 |
| # Warmup epochs | 50 | 5 | 5 | 5 | 5 |
| Weight decay | 1e-5 | 1e-5 | 1e-5 | 1e-5 | 1e-5 |
| $K$ | 1 | 1 | 1 | 1 | 1 |
| Coarsening algorithm | Newman | Louvain | Louvain | Loukas ($\alpha = 0.1$) | Louvain |
| # Parameters | 437,389 | 124,565 | 121,913 | 352,695 | 517,446 |

Table 10: Hyperparameters of GraphGPS + HDSE for two LRGB datasets from [23].

| Hyperparameter | Peptides-func | Peptides-struct |
|---|---|---|
| # GPS Layers | 4 | 4 |
| Hidden dim | 96 | 96 |
| GPS-MPNN | GatedGCN | GatedGCN |
| GPS-GlobAttn | Transformer | Transformer |
| # Heads | 4 | 4 |
| Attention dropout | 0.5 | 0.5 |
| Graph pooling | mean | mean |
| Positional Encoding | LapPE-10 | LapPE-10 |
| PE dim | 16 | 16 |
| PE encoder | DeepSet | DeepSet |
| Batch size | 16 | 128 |
| Learning Rate | 0.0003 | 0.0003 |
| # Epochs | 200 | 200 |
| # Warmup epochs | 5 | 5 |
| Weight decay | 0 | 0 |
| $K$ | 0 | 1 |
| Coarsening algorithm | Newman | METIS ($\alpha = 0.1$) |
| # Parameters | 505,866 | 506,235 |

we have covered in the setting part of the experiment section that datasets share in common, we list other settings in Table 11. It's important to note that our hyperparameters were determined within the SGFormer's grid search space. Furthermore, all other experimental parameters, including dropout, batch size, training schemes, optimizer, etc., are consistent with those used in the SGFormer [79]. The testing accuracy achieved by the model that reports the highest result on the validation set is used for evaluation. And each experiment is repeated 10 times to get the mean value and error bar.

**SGFormer on Graph-level Tasks.** To accurately demonstrate the capabilities of SGFormer on these datasets, we use all the same experimental settings and conduct the same grid search as outlined in GraphGPS [68].

Table 11: GOAT + HDSE dataset-specific hyperparameter settings.

| Dataset | $K$ | $|V^1|$ | Hidden dim | # Heads | # Glob. Layers | Local GNN | # GNN Layers | # Epochs | LR |
|---------|-----|---------|-----------|---------|----------------|-----------|--------------|----------|-----|
| Cora | 1 | 32 | 128 | 4 | 1 | GCN | 2 | 500 | 1e-2 |
| Citeseer | 1 | 200 | 128 | 2 | 1 | GCN | 2 | 500 | 1e-2 |
| Pubmed | 1 | 64 | 128 | 1 | 1 | GCN | 2 | 500 | 1e-2 |
| Actor | 1 | 200 | 128 | 2 | 1 | GCN | 2 | 1000 | 1e-2 |
| Squirrel | 1 | 128 | 128 | 1 | 3 | GCN | 2 | 500 | 1e-2 |
| Chameleon | 1 | 32 | 128 | 1 | 3 | GCN | 2 | 500 | 1e-2 |
| ogbn-proteins | 1 | 1024 | 128 | 2 | 1 | GraphSAGE | 4 | 1000 | 5e-4 |
| ogbn-arxiv | 1 | 1024 | 256 | 1 | 1 | GCN | 7 | 2000 | 5e-4 |
| arxiv-year | 1 | 2048 | 128 | 4 | 1 | GAT | 1 | 500 | 1e-3 |
| ogbn-products | 2 | 1024 | 256 | 4 | 1 | GraphSAGE | 5 | 1000 | 3e-3 |
| ogbn-100M | 1 | 1024 | 256 | 1 | 1 | GCN | 3 | 50 | 1e-3 |

# C  Additional Experimental Results

## C.1  Performance on Large-Scale Heterophilic Graphs

We run additional experiments on the Pokec and snap-patents datasets [55]. Pokec represents the friendship network of a Slovak online social platform, where nodes correspond to users and directed edges denote friendship relationships. Each node is labeled with the user's reported gender, and features are derived from profile information, including geographical region, registration time, and age. The snap-patents dataset comprises utility patents in the United States, where nodes represent individual patents and edges indicate citation relationships between them. Node features are extracted from patent metadata.

We use the default splits and features from the LINKX [55] and reported the mean accuracy over 5 runs. The results further demonstrate the effectiveness of our HDSE on large-scale heterophilic graphs.

Table 12: Performance on large-scale heterophilic graphs.

| | Pokec Accuracy↑ | snap-patents Accuracy↑ |
|---|---|---|
| LINKX | 82.04 ± 0.07 | 61.95 ± 0.12 |
| GOAT | 84.69 ± 0.18 | 62.43 ± 0.37 |
| **GOAT + HDSE** | 85.88 ± 0.33 | 63.56 ± 0.26 |

## C.2  Sensitivity Analysis of Maximal Hierarchy Level $K$

We conduct a sensitivity analysis on the maximum hierarchy level $K$ on ZINC. The results are shown in Table 13. Note that when $K = 0$, HDSE degenerates into SPD, leading to a worse performance. This result of $K$ is about graph classification tasks, where the size of graphs is typically small. Therefore, at level 1 ($K = 1$), the quantity of coarsened nodes is quite small, eliminating the necessity for a higher $K$. We further investigate the impact of $K$ on large-graph node classification, across 3 datasets: Squirrel, arxiv-year, and ogbn-arxiv. Based on the results displayed in Table 14, we can make the following observations: (1) $K = 1$ does not consistently yield the best results. Optimal performance is achieved with $K = 2$ on some datasets. (2) The improvement brought about by $K = 2$ over $K = 1$ is relatively minor.

The variation in the optimal $K$ could stem from the distinct hierarchical structures inherent in different graphs. Larger graphs may possess more pronounced multi-level structures, thus necessitating a higher $K$. However, the slight improvement resulting from a larger $K$ could suggest limitations in the coarsening algorithm.

This study reinforces our selection of $K = 1$, aligning with results from other hierarchical graph transformer papers such as HSGT [96] and ANS-GT [91]. We anticipate that the real potential of a higher $K$ will be revealed through the application of a proper, effective coarsening algorithm on graphs with hierarchical community structures. We look forward to exploring this in the future.

Table 13: Sensitivity analysis on the maximum hierarchy level $K$ of GraphGPS + HDSE on ZINC.

|  | $K = 0$ (SPD) | $K = 1$ | $K = 2$ |
|---|---|---|---|
| ZINC ↓ | 0.069 ± 0.003 | 0.062 ± 0.003 | 0.064 ± 0.004 |

Table 14: Sensitivity analysis on the maximum hierarchy level $K$ of GOAT + HDSE.

|  | Squirrel ↑ | arxiv-year ↑ | ogbn-arxiv ↑ |
|---|---|---|---|
| $K = 1$ | 43.2 ± 2.4 | 54.23 ± 0.26 | 73.26 ± 0.19 |
| $K = 2$ | 43.8 ± 2.1 | 54.51 ± 0.17 | 73.36 ± 0.15 |

## C.3 Sensitivity Analysis of Maximum Distance Length $L$

Table 15: Overview of the graph diameters of datasets used in graph classification

|  | ZINC | MNIST | CIFAR10 | PATTERN | CLUSTER | Peptides-func | Peptides-struct |
|---|---|---|---|---|---|---|---|
| Average Diameter | 12.47 | 6.85 | 9.17 | 2.00 | 2.17 | 56.86 | 56.86 |
| Maximum Diameter | 22 | 8 | 12 | 3 | 3 | 159 | 159 |

Table 16: Sensitivity analysis on the maximum distance length $L$.

|  | Peptides-func ↑ | Peptides-struct ↑ |
|---|---|---|
| GraphGPS + HDSE ($L = 20$) | 0.7105 ± 0.0051 | 0.2481 ± 0.0016 |
| GraphGPS + HDSE ($L = 30$) | 0.7156 ± 0.0058 | 0.2457 ± 0.0013 |
| GraphGPS + HDSE ($L = 50$) | 0.7124 ± 0.0053 | 0.2466 ± 0.0021 |

For each graph classification dataset, we calculate the graph diameter of each graph in the dataset and then compute the average graph diameter and maximum graph diameter for the entire dataset, as detailed in Table 15. Note that we use high-level HDSE to deal with node classification on large graphs; therefore, we do not calculate the distances between the nodes in these large graphs. The data indicates $L = 30$ is a reasonable choice, as it encompasses most of the graph diameters. We do not use a larger number as we hypothesize that for graphs with larger diameters, the utility of detailed information loses significance beyond a certain distance.

Additionally, we conducted a sensitivity analysis regarding the selection of $L$, as outlined in Table 16, which confirms that $L = 30$ is an appropriate choice.

## C.4 Coarsening Runtime

Table 17 gives the runtime of coarsening algorithms (including distance calculation) on graph-level tasks, illustrating the practicality of our method. The Newman algorithm is unsuited for larger graphs due to high complexity. In addition, our HDSE module almost does not increase the runtime of the baselines. For example, GraphGPS runs at 10 seconds per epoch, compared to 11 seconds per epoch with HDSE module on ZINC.

Additionally, for all large-scale graphs, we employ METIS due to its efficiency with a time complexity of $O(|E|)$. This makes it highly effective for partitioning large graphs, such as ogbn-products, in less than 5 minutes, and even the vast ogbn-papers100M, with a size of 0.1 billion nodes, requires only 59 minutes.

Table 17: Empirical runtime of coarsening algorithms.

| Algorithm | ZINC | PATTERN | MNIST | P-func |
|---|---|---|---|---|
| METIS | 31s | 0.1h | 0.2h | 0.1h |
| Newman | 88s | >500h | 18h | 1.6h |
| Louvain | 76s | 5h | 1.6h | 1.1h |

## C.5 Impact of Coarsening Algorithms on Large-scale Graphs

Table 18: Node classification results with linear coarsening algorithms on Cora, CiteSeer, and PubMed.

|  | Cora ↑ | CiteSeer ↑ | PubMed ↑ |
|---|---|---|---|
| GOAT | 82.1 ± 0.9 | 71.6 ± 1.3 | 78.9 ± 1.5 |
| **GOAT + HDSE** (METIS) | 83.9 ± 0.7 | 73.1 ± 0.7 | 80.6 ± 1.0 |
| **GOAT + HDSE** (Loukas) | 83.5 ± 0.9 | 72.5 ± 0.6 | 79.8 ± 0.9 |

Our study on coarsening algorithms in Table 4 focuses on the ZINC dataset, where the size of graphs is typically small (around 20 nodes). The Newman algorithm exhibits optimal performance on these small graphs; however, on large-scale graphs, we use a linear complexity algorithm METIS.

To further assess the impact of linear coarsening algorithms, we conduct additional experiments to study the impact of linear coarsening algorithms on node classification across three datasets: Cora, CiteSeer, and PubMed. The results, as shown in Table 18, demonstrate the advantage of METIS, which is the coarsening algorithm used for node classification in our experiments.

## C.6 Synthetic Community Dataset

We evaluate the **Community-small** dataset from GraphRNN [88], a synthetic dataset featuring community structures. It comprises 100 graphs, each with two distinct communities. These communities are generated using the Erdos-Renyi model (E-R). Node features are generated from random numbers and node labels are determined by their respective cluster numbers with accuracy as the chosen evaluation metric. We use the a random train/validation/test split ratio of 60%/20%/20%.

Table 19: Node classification on synthetic community datasets.

| Dataset | GT | GT + SPD | GT + HDSE |
|---|---|---|---|
| Community-small | 64.7 ± 1.1 | 81.5 ± 1.7 | 88.6 ± 0.9 |

We select the Louvain method as our coarsening algorithm and integrate the HDSE module into the Graph Transformer (GT). As shown in Table 19, the GT struggles to detect such structures; and solely utilizing SPD proves inadequate; however, our HDSE, enriched with coarsening structural information, effectively captures these structures.

## C.7 ANS-GT + HDSE

We validate the performance of our HDSE framework using the efficient ANS-GT [91], which uses a multi-armed bandit algorithm to adaptively sample nodes for attention. We use the Louvain method as our coarsening algorithm. And for each pair of nodes sampled adaptively by the ANS-GT, we calculate their HDSE and bias the attention computation. For fair comparisons, we tune the hyperparameters using the same grid search as reported in their paper [91]. Note that we report the supervised learning setting (different from the text), since this is the one considered in the ANS-GT [91]. Overall, Table 20 shows that HDSE yields consistent performance improvements, even in this challenging scenario, where nodes are sampled.

## C.8 Gapformer + HDSE

To further validate the effectiveness of our HDSE framework, we also integrate our high-leve HDSE with Gapformer [56], and observe promising results, as reported in Table 21.

Note that we follow the supervised split setting (48%/32%/20% training/validation/test sets) used in the Gapformer [56] here.

## C.9 Clustering Coefficients Analysis

We check if there is a correlation with the cluster structure according to [35], by computing clustering coefficients on five benchmarks in Table 22, but we do not observe a direct correlation. Notably, the

Table 20: **Node classification accuracy** on ANS-GT + HDSE (%). The baseline results were taken from [91]. We apply 3 runs on random data splitting. $^+$ indicates the results obtained from our re-running.

| Model | Cora ↑ | Citeseer ↑ | Pubmed ↑ |
|---|---|---|---|
| GCN | 87.33 ± 0.38 | 79.43 ± 0.26 | 84.86 ± 0.19 |
| GAT | 86.29 ± 0.53 | **80.13 ± 0.62** | 84.40 ± 0.05 |
| APPNP | 87.15 ± 0.43 | 79.33 ± 0.35 | 87.04 ± 0.17 |
| JKNet | 87.70 ± 0.65 | 78.43 ± 0.31 | 87.64 ± 0.26 |
| H2GCN | 87.92 ± 0.82 | 77.60 ± 0.76 | 89.55 ± 0.14 |
| GPRGNN | 88.27 ± 0.40 | 78.46 ± 0.88 | 89.38 ± 0.43 |
| GT | 71.84 ± 0.62 | 67.38 ± 0.76 | 82.11 ± 0.39 |
| SAN | 74.02 ± 1.01 | 70.64 ± 0.97 | 86.22 ± 0.43 |
| Graphormer | 72.85 ± 0.76 | 66.21 ± 0.83 | 82.76 ± 0.24 |
| Gophormer | 87.65 ± 0.20 | 76.43 ± 0.78 | 88.33 ± 0.44 |
| Coarformer | 88.69 ± 0.82 | 79.20 ± 0.89 | 89.75 ± 0.31 |
| ANS-GT | 88.60 ± 0.45 | 77.75 ± 0.79$^+$ | 89.56 ± 0.55 |
| **ANS-GT + HDSE** | **89.67 ± 0.39** | 78.31 ± 0.58 | **90.63 ± 0.26** |

Table 21: Node classification results of Gapformer with and without HDSE on Cora, CiteSeer, and PubMed.

| | Cora ↑ | CiteSeer ↑ | PubMed ↑ |
|---|---|---|---|
| Gapformer | 87.3 ± 0.7 | 76.2 ± 1.4 | 88.9 ± 0.4 |
| **Gapformer + HDSE** | 88.4 ± 0.7 | 76.9 ± 0.6 | 89.7 ± 0.5 |

ZINC dataset, which comprises small molecules, has a low clustering coefficient; however, our HDSE shows a significant improvement on it. This improvement could be attributed to the HDSE capturing chemical motifs that cannot be captured by the clustering coefficient, as illustrated in Figure 2.

Table 22: Clustering Coefficients Analysis.

| Model | ZINC MAE ↓ | MNIST Accuracy ↑ | CIFAR10 Accuracy ↑ | PATTERN Accuracy ↑ | CLUSTER Accuracy ↑ |
|---|---|---|---|---|---|
| Average Clust. Coeff. | 0.006 | 0.477 | 0.454 | 0.427 | 0.316 |
| GT | 0.226 ± 0.014 | 90.831 ± 0.161 | 59.753 ± 0.293 | 84.808 ± 0.068 | 73.169 ± 0.622 |
| **GT + HDSE** | 0.159 ± 0.006 | 94.394 ± 0.177 | 64.651 ± 0.591 | 86.713 ± 0.049 | 74.223 ± 0.573 |
| SAT | 0.094 ± 0.008 | – | – | 86.848 ± 0.037 | 77.856 ± 0.104 |
| **SAT + HDSE** | 0.084 ± 0.003 | – | – | 86.933 ± 0.039 | 78.513 ± 0.097 |
| GraphGPS | 0.070 ± 0.004 | 98.051 ± 0.126 | 72.298 ± 0.356 | 86.685 ± 0.059 | 78.016 ± 0.180 |
| **GraphGPS + HDSE** | 0.062 ± 0.003 | 98.367 ± 0.106 | 76.180 ± 0.277 | 86.737 ± 0.055 | 78.498 ± 0.121 |

# D  HDSE Visualization

Here, we demonstrate that our HDSE method also provides interpretability compared to the classic GT. We train the GT + HDSE and GT on ZINC and Peptides-func graphs, and compare the attention scores between the selected node and other nodes. Figure 5 visualizes the attention scores on ZINC and Peptides-func. The results indicate that, after integrating the HDSE bias, the attention mechanism tends to focus on certain community structures rather than individual nodes as seen in classic attention. Furthermore, different selected nodes lead to different attention weights and consistently demonstrate our HDSE's capability to capture a multi-level hierarchical structure.

# E  Additional Discussion

**Positional Encoding or Structural Encoding?**

We would like to clarify that our HDSE method aligns with the definitions of structural encoding. While GraphGPS [68] does classify SPD encoding under relative positional encoding, it defines

relative structural encoding as "allow two nodes to understand how much their structures differ". Given that our HDSE not only incorporates SPD information but also embeds multi-level graph hierarchical structures, it is reasonable to classify it under the category of structural encoding.

**Limitations.** In larger graphs, the presence of multi-level structures may require a higher maximal hierarchy level, $K$. The marginal improvements observed with increased $K$ may indicate limitations in the coarsening algorithm. We anticipate that the real potential of a higher $K$ will be revealed through the application of a proper, effective coarsening algorithm on graphs with hierarchical community structures. We look forward to exploring this in the future.

**Broader Impacts.** This paper presents work whose goal is to advance the field of Machine Learning. There are many potential societal consequences of our work, none which we feel must be specifically highlighted here.

# F   Further Related Works

**Graph Transformers over Clustering Pooling.** [45] employs a hybrid approach that integrates a neighbor-sampling local module with a global module, the latter featuring a trainable, fixed-size codebook obtained by K-Means to represent global centroids, which is noted for its efficiency. Meanwhile, Gapformer [56] involves the incorporation of a graph pooling layer designed to refine the key and value matrices into pooled key and value vectors through graph pooling operations. This approach aims to minimize the presence of irrelevant nodes and reduce computational demands. However, the performance of these methods remains constrained due to a lack of effective inductive biases.

**Graph Transformers over Virtual Nodes.** Several graph transformer models utilize anchor nodes or virtual nodes for message propagation. For instance, Graphormer [86] introduces a virtual node and establishes connections between the virtual node and each individual node. AGFormer [41] selects representative anchors and transforms node-to-node message passing into an anchor-to-anchor and anchor-to-node message passing process. Additionally, AGT [62] extracts structural patterns from subgraph views and designs an adaptive transformer block to dynamically integrate attention scores in a node-specific manner.

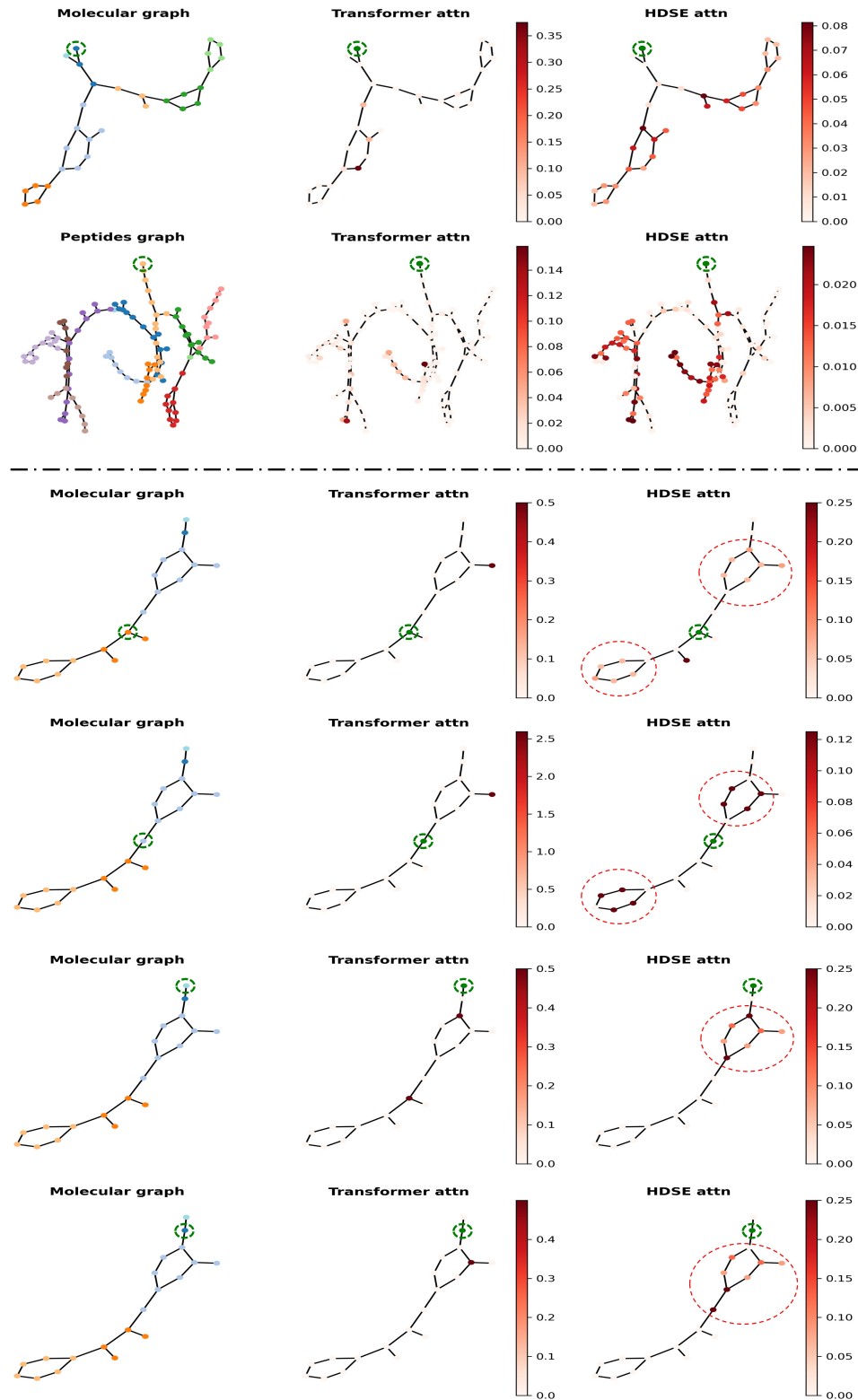

Figure 5: Visualization of attention weights for the transformer attention and HDSE attention. The left side illustrates the graph coarsening result. The center column displays the attention weights of a randomly sample node (enclosed in a green dashed box) learned by the classic GT, while the right column showcases the attention weights learned by the HDSE attention. Note that different randomly selected nodes consistently demonstrate the ability to capture a multi-level hierarchical structure.

