# OpenReview forum: "Enhancing Graph Transformers with Hierarchical Distance Structural Encoding"
_NeurIPS.cc/2024/Conference — NeurIPS 2024 poster_

### Official Review · Reviewer_o879 · 2024-07-07

**Soundness:** 3
**Presentation:** 4
**Contribution:** 3
**Rating:** 7
**Confidence:** 3

**Summary:**

This work proposes a new structural encoding for graphs, which can be applied in e.g. graph transformers. The method relies on some graph partitioning / coarsening methods to generate hierarchical clustering of a graph, and the structural encoding is more informative than shortest path structural encoding. The paper proves the the new structural encoding has higher expressivity, and empirical results are good on most graph datasets.

**Strengths:**

The method is novel, instead of running a graph transformer on different hierarchical clusters, it makes use of the shortest path on different hierarchies.
Overall the writing is pretty good, the narrative is clear and easily understandable. The illustration is also nice.
The theoretical and empirical results are strong. The experiments are exhaustive.

**Weaknesses:**

The graph partitioning / coarsening preprocess, including the paritioning algorithm and shortest path computation, can also be complex.

**Questions:**

According to the definition of edge set $E^{k+1}$ in line 78, wouldn't there be information loss? If there are multiple edges across two clusters, the number of edges is not represented.
In section 3.3, is the partition matrix low rank? Especially if the coarsening ratio is close to 1, I guess the partition matrix would be almost full rank.

**Limitations:**

According to authors, for larger graphs, more hierarchies are required.

---

> ### Author Rebuttal · Authors · 2024-08-07
>
> **Thank you for clearly understanding the core of our work and fully recognizing our contributions!**
>
> **W1: Complexity**
>
> >The graph partitioning / coarsening preprocess, including the paritioning algorithm and shortest path computation, can also be complex.
>
> This is a fair point. We acknowledge that our approach introduces an additional cost. However, we would like to clarify that this cost is manageable. In our configuration, we employ the METIS algorithm for coarsening on large graphs, which is characterized by a linear time complexity of $O(|E|)$, making it efficient for partitioning graphs. For example, as reported in in *Appendix C.4* of our manuscript, it takes less than five minutes for coarsening and shortest path computation on the ogbn-products graph with 2 million nodes.
>
> We believe that the enhanced performance of our method, as demonstrated in our empirical evaluations, justifies the additional small computational overhead.
>
>
> **Q1: Theoretical Details**
>
> > According to the definition of edge set Ek+1 in line 78, wouldn't there be information loss? If there are multiple edges across two clusters, the number of edges is not represented.
>
> Thank you for carefully checking the theoretical details! We adhere to the standard definitions of Graph Hierarchies. Under this framework, some information loss regarding the number of edges between clusters is inevitable. However, our HDSE framework addresses this by retaining the original edge information within $\text{GHD}^0$, the shortest path distance matrix of the input graph.
>
> > In section 3.3, is the partition matrix low rank? Especially if the coarsening ratio is close to 1, I guess the partition matrix would be almost full rank.
>
> Yes, the partition matrix is inherently low-rank. In our settings, the coarsening ratio is much smaller than 1, ensuring that the partition matrix remains low-rank.
>
> Once again, we sincerely appreciate your acknowledgment of our work.

---

> > ### Comment · Reviewer_o879 · 2024-08-11
> >
> > Thank you for your feedback!

---

> > > ### Author Response · Authors · 2024-08-11
> > >
> > > Dear Reviewer o879,
> > >
> > > This final confirmation is highly appreciated!
> > >
> > > Best regards,
> > >
> > > Authors

---

### Official Review · Reviewer_4HNX · 2024-07-18

**Soundness:** 3
**Presentation:** 3
**Contribution:** 3
**Rating:** 6
**Confidence:** 4

**Summary:**

This paper leverages graph coarsening techniques to help the graph transformer capture hierarchical distance information on a graph, improving its performance on both node-level and graph-level tasks. Besides empirical validation, the paper also provides theoretical guarantee about the better expressiveness and generalization of the graph transformer with the proposed HDSE.

**Strengths:**

- This paper is well organized and clearly written.
- The method incorporates coarsening technology, integrating distance information at different hierarchical levels into the attention score. It is easy to understand and implement and is somewhat novel. Additionally, it has a certain theoretical guarantee regarding the expressiveness and generalization of the proposed method.
- The experiments are comprehensive, evaluating the model's performance on homophilic and heterophilic graphs, as well as on large-scale graphs. Efficiency and ablation studies are also conducted, along with visualizations. The improvements are significant on some datasets, such as Peptides-func, Chameleon, and Squirrel.

**Weaknesses:**

- There are some missing results in the experiments (mainly in Tables 2 and 5). For example, the results for models such as ANS-GT, NAGphormer, and LINKX on Actor, Squirrel, and Chameleon are missing. Although implementing all work on all datasets is time-consuming, the missing results reduce the paper's convincingness. Additionally, some results differ from previous reports. For instance, LINKX reported its performance as about 56 on the arxiv-year dataset [1], but in this paper, it is around 53.53. Is this because of different experimental settings?
- Besides, the performance of the model on large-scale heterophilic graphs remains unknown. The few heterophilic graphs shown in Table 5 are not large. To further demonstrate the model's performance on large-scale heterophilic datasets, experiments on datasets such as pokec, snap-patents, and wiki [1] are recommended. This would enhance the quality of the experiments.

[1] Large Scale Learning on Non-Homophilous Graphs: New Benchmarks and Strong Simple Methods

**Questions:**

Besides the questions mentioned above, there are some additional questions:

**Q1:** In Equation 7 and the illustration below, could you please explain why $\text{GHD}^m\prod \limits_{l=0}^{c-1}P^l$  computes distances from input nodes to clusters at the c-level graph hierarchy? For the input node, why is it not $\text{GHD}^0$? $\text{GHD}^m$ represents the distance at hierarchy $m$. What is the meaning of $\text{GHD}^m$ with the product of projection (coarsening) matrices from level $0$ to $c-1$ (i.e. $\prod \limits_{l=0}^{c-1}P^l$)?

**Q2:** In Table 6, the efficiency of GOAT+HDSE, SGFormer, and NodeFormer was compared. However, since NodeFormer and SGFormer use linear attention, is GOAT+HDSE still competitive in terms of efficiency when the graphs are larger (i.e., ogbn-products and ogbn-papers100m)? As the method is claimed to have scalability, illustrations on such large-scale graphs would be beneficial. Additionally, as HDSE is based on graph coarsening and GHD, the cost of coarsening and computation of GHD at each level should also be considered, especially on graphs with more nodes than those used in Table 17.

**Q3:** In the Experiment Setting, the same hyperparameters are used for the proposed method and the baseline transformers. It is also claimed that the hyperparameters are determined within SGFormer's grid search space. However, the optimal hyperparameters for the proposed method may not be suitable for the baseline models. Could you clarify this?

**Q4:** In Appendix 4, the attention scores are displayed between the selected node and the other nodes. How was the node selected? Would different selected nodes lead to different attention weights and still reflect the capability to capture a multi-level hierarchical structure?

**Limitations:**

When graphs are larger, a higher maximal hierarchy level $K$, is needed to obtain multi-level structure information, which may be limited by current coarsening algorithms. This limitation can be addressed with more effective coarsening algorithms.

---

> ### Author Rebuttal · Authors · 2024-08-07
>
> **We greatly appreciate the very detailed feedback and your recognition of our contributions! We hope our response below will further enhance your confidence in our work.**
>
> **W1: Incompleteness of Experimental Results**
>
> > The results for models such as ANS-GT, NAGphormer, and LINKX on Actor, Squirrel, and Chameleon are missing.
>
> Since we experimented with well-known datasets, **we reported the baseline results from their respective original papers**. We did not run them ourselves, but are happy to add results based on your recommendations. We have run all the missing results you mentioned. The additional results below further demonstrate the effectiveness of our HDSE. We will include these results in the revised version.
>
> |                 | Actor↑         | Squirrel↑      | Chameleon↑     |
> | --------------- | -------------- | -------------- | -------------- |
> | ANS-GT          | 35.2 ± 1.3     | 40.8 ± 2.1     | 42.6 ± 2.7     |
> | NAGphormer      | 34.3 ± 0.9     | 39.7 ± 0.8     | 40.3 ± 1.7     |
> | LINKX           | 36.1 ± 1.5     | 41.9 ± 1.2     | 43.8 ± 2.9     |
> | **GOAT + HDSE** | **38.0 ± 1.5** | **43.2 ± 2.4** | **46.0 ± 3.2** |
>
> > The arxiv-year results differ from previous reports.
>
> Yes, the observed discrepancies on the arxiv-year dataset are due to different experimental settings. We employed the experimental setup used in GOAT to ensure a fair comparison. The results we reported—53.53 for LINKX and 53.57 for GOAT—are consistent with those in the original GOAT paper (Table 2 in GOAT paper).
>
> **W2: Performance on Large-Scale Heterophilic Graphs**
>
> >  Experiments on datasets such as pokec, snap-patents, and wiki are recommended.
>
> We appreciate your suggestion.
>
> We ran additional experiments on the pokec and snap-patents datasets. We attempted to run experiments on the wiki dataset, but the download link for the label file 'wiki_views2M' was no longer valid. We contacted the authors of the LINKX paper but haven't received a response yet. We utilized the default splits and features from the LINKX paper and reported the mean accuracy over 5 runs. The results further demonstrate the effectiveness of our HDSE on large-scale heterophilic graphs. We will include the results in the revised version.
>
> |                 | pokec↑           | snap-patents↑    |
> | --------------- | ---------------- | ---------------- |
> | LINKX           | 82.04 ± 0.07     | 61.95 ± 0.12     |
> | GOAT            | 84.69 ± 0.18     | 62.43 ± 0.37     |
> | **GOAT + HDSE** | **85.88 ± 0.33** | **63.56 ± 0.26** |
>
> **Q1: Explanation of GHD Computation**
>
> > Why $\mathrm{GHD}^m (\prod_{l=0}^{c-1} P^l)$ computes distances from input nodes to clusters at the $c$-level graph hierarchy?
>
> We apologize for the confusion.
>
> As defined in Eq.2, $\mathrm{GHD}^m \in \mathbb{R}^{|V| \times |V|}$ represents **the shortest path distance between any two *input nodes* at the $m$-level graph hierarchy.** $\forall m, \mathrm{GHD}^m$ has the same size as $\mathrm{GHD}^0$ (see illustration in Figure 1).
>
> In Eq.7, our high-level HDSE computes, at each level $c\leq m \leq K$, distances between input nodes and clusters obtained by coarsening (i.e., super nodes at the $c$-level graph hierarchy). This is achieved by multiplying the projection matrices $\prod_{l=0}^{c-1} P^l$ to $\mathrm{GHD}^m$. In effect, it is equvalent to selecting corresponding columns from $\mathrm{GHD}^m$. For instance, referring to Figure 1, $\mathrm{GHD}^1P^0 \in \mathbb{R}^{11 \times 3}$ calculates the distances from input nodes to the super nodes at $1$-level graph hierarchy, essentially selecting the first, fourth, and tenth columns from $\mathrm{GHD}^1$.
>
> Likewise, $\mathrm{GHD}^m  (\prod_{l=0}^{c-1} P^l) \in \mathbb{R}^{|V| \times |V^c|}$ selects $|V^c|$ columns from $\mathrm{GHD}^m$ to represent the distances, at the $m$-level graph hierarchy, between the input nodes and the $c$-level super nodes (i.e., clusters obtained through coarsening).
>
> **Q2: Efficiency on Large-Scale Graphs**
>
> > Is GOAT+HDSE still competitive in terms of efficiency when the graphs are larger (i.e., ogbn-products and ogbn-papers100M)?
>
> Thank you for the suggestion. As noted in lines 214-218, GOAT also uses linear attention. The integration of HDSE does not increase the complexity of GOAT. We provide additional results on larger graphs below, further demonstrating the competitiveness of GOAT+HDSE in terms of efficiency.
>
> | Training time per epoch | ogbn-products | ogbn-papers100M |
> | ----------------------- | ------------- | --------------- |
> | NodeFormer              | 5.6s          | 595.1s          |
> | SGFormer                | **4.8s**      | 579.4s          |
> | **GOAT + HDSE**         | 5.3s          | **446.5s**      |
>
> We have reported the empirical runtime for coarsening and the computation of GHD on ogbn-products (5min using METIS) and ogbn-papers100m (59min using METIS) in *Appendix C.4* of our manuscript.  We will add the results in Table 17.
>
> **Q3:  Optimal Hyperparameters for Baselines**
>
> > Could you clarify the optimal hyperparameters for the baselines?
>
> We apologize for the confusion.
>
> It's important to note that **we obtained the results for all baselines from their original papers or established leaderboards, where they were optimally tuned**.
>
> Please also note that we adopted two distinct experimental setups:
>
> 1. For **Graph-Level Tasks**, we used **Base Model + HDSE**. The base model (e.g., GT, SAT, GraphGPS, or GRIT) was optimally tuned according to the original paper. ***We used the same hyperparameters as the base model to demonstrate the plug-and-play capability of our HDSE***.
>
> 2. For **Large-Graph Node Classification**, our model is **GOAT + HDSE**. We tuned the hyperparameters of our model within SGFormer's grid search space.
>
> We will further clarify that in the revised version.
>
> **Q4: Selection of Nodes to Display Attention Scores**
>
> Thank you for your careful reading. Due to space constraints, please refer to the "Global Reply": #G1 item.

---

> > ### Comment · Reviewer_4HNX · 2024-08-08
> >
> > Thanks for your detailed feedback, which has addressed some of my concerns. I have raised my scores for Soundness and Rating.

---

> > > ### Author Response · Authors · 2024-08-09
> > >
> > > Dear Reviewer 4HNX,
> > >
> > > Thank you very much for taking the time to review our rebuttal! We highly appreciate your positive and insightful assessment of our work.
> > >
> > > In case there are any additional concerns we can address, please let us know.
> > >
> > > Best regards,
> > >
> > > Authors

---

### Official Review · Reviewer_p468 · 2024-07-18

**Soundness:** 3
**Presentation:** 3
**Contribution:** 2
**Rating:** 6
**Confidence:** 3

**Summary:**

To enhance the effectiveness and scalability of graph transformers, this paper proposes a hierarchical distance structural encoding (HDSE) method to incorporate hierarchical structural information with graph transformers. Theoretical analysis of graph transformer equipped with HDSE shows the improvements of both expressiveness and generalization. Experiments have been conducted to verify the effectiveness and efficiency of the proposed method.

**Strengths:**

1. It is interesting and reasonable to improve the performance of graph transformer based on the hierarchical structural information of the input graph. Moreover, the proposed method is easy to be applied on existing graph transformer models.
2. Theoretical analysis are provided to guarantee the improvements of expressiveness and generalization of combining HDSE with graph transformer.
3. A technique of speeding up HDSE to large-scale graphs is provided and verified in experiments on large graphs with millions of nodes.
4. The paper is well-organized and describes the motivation and methodology clearly.

**Weaknesses:**

1. It can be seen from Table 4 that the performance of the proposed method varies with different coarsening algorithms and transformer backbones, which should be discussed in details.

2. The experimental settings are not described clearly. Four model + HDSE methods are used for graph-level tasks but only one GOAT + HDSE used for node classification, which makes the results slightly inconvincible.

**Questions:**

1. Although combining the hierarchical structural information into graph transformer indeed improves the performance of the transformer, the improvements vary with different coarsening algorithms. Thus, can the authors provide theoretical analysis on what is the optimal hierarchical structural information and conduct more experiments on how to select suitable coarsening algorithm in practice?

2. The authors integrate HDSE into GT, SAT, GraphGPS and GRIT for graph classification and regression experiments (in Table 2), but only use GOAT + HDSE in the node classification tests (in Table 5), why? Moreover, the results of node classification on classical datasets Core, CiteSeer and PubMed should be reported and discussed in the Evaluation section rather than Appendix.

**Limitations:**

Please refer to weakness and questions.

---

> ### Author Rebuttal · Authors · 2024-08-07
>
> **We greatly appreciate your comprehensive understanding of our work and recognizing our contributions! We hope our response below will further enhance your confidence in our work.**
>
> **W1 & Q1: Impact of Coarsening Algorithms**
>
> >It can be seen from Table 4 that the performance of the proposed method varies with different coarsening algorithms and transformer backbones, which should be discussed in details.
>
> >Can the authors provide theoretical analysis on what is the optimal hierarchical structural information and conduct more experiments on how to select suitable coarsening algorithm in practice?
>
> Thank you for the insightful comment and question.
>
>
>
> Our study on coarsening algorithms in Table 4 focuses on the ZINC dataset, where the size of graphs is typically small (around 20 nodes). The Newman algorithm exhibits optimal performance on these small graphs; however, as delineated in Appendix C.4, its high computational complexity makes it impractical for larger graphs. Therefore, **for coarsening on large graphs, we recommend using a *linear complexity* algorithm, such as METIS or Loukas**.
>
> Following your suggestion, we conducted additional experiments to study the impact of *linear coarsening algorithms* on node classification across three datasets: Cora, CiteSeer, and PubMed. The results, as shown below, demonstrates the advantage of METIS, which is the coarsening algorithm used for node classification in our experiments.
>
> |                      | Cora↑          | CiteSeer↑      | PubMed↑        |
> | -------------------- | -------------- | -------------- | -------------- |
> | GOAT                 | 82.1 ± 0.9     | 71.6 ± 1.3     | 78.9 ± 1.5     |
> | GOAT + HDSE (METIS)  | **83.9 ± 0.7** | **73.1 ± 0.7** | **80.6 ± 1.0** |
> | GOAT + HDSE (Loukas) | 83.5 ± 0.9     | 72.5 ± 0.6     | 79.8 ± 0.9     |
>
> Based on our observations, **we suggest using higher-complexity algorithms like Newman or Louvain for small graphs, and linear-complexity algorithms like METIS for large graphs**. We will include the discussion in the revised manuscript.
>
> Additionally, different coarsening algorithms tend to produce different hierarchical structures, which benefit different types of graph classification tasks. For instance, preserving rings is crucial for certain tasks. Similarly, the Newman algorithm, which calculates edge betweenness and tends to identify bridges, may be particularly well-suited to networks where bridges play a critical role.
>
> Hence, we now treat the choice of coarsening algorithm as a hyperparameter, considering the unique structures inherent to different graphs and the broad scope of various applications. This allows graph experts to tailor their analysis by selecting the most appropriate coarsening algorithm for the structures of specific graphs.
>
> Further theoretical exploration into the optimal hierarchical structural information for different types of graphs is very interesting. We look forward to exploring this in the future.
>
>
>
> **W2 & Q2: Clarity of Experimental Settings**
>
> > The experimental settings are not described clearly. Four model + HDSE methods are used for graph-level tasks but only one GOAT + HDSE used for node classification, which makes the results slightly inconvincible.
>
> >The authors integrate HDSE into GT, SAT, GraphGPS and GRIT for graph classification and regression experiments (in Table 2), but only use GOAT + HDSE in the node classification tests (in Table 5), why?
>
> We apologize for the confusion.
>
> Please note that we have two separate experimental settings: one for **graph-level tasks using Conventional Graph Transformers + HDSE**, another for **large-graph node classification using *Linear-attention* Graph Transformer (e.g., GOAT) + HDSE** (high-level HDSE). This distinction arises from the constraints imposed by the quadratic complexity of conventional graph transformers (like GT, SAT, GraphGPS, GRIT), which are not feasible for large-scale graphs due to out-of-memory issues.
>
> Therefore, since **large-graph node classification necessitates the use of Linformer-style linear-attention graph transformers such as GOAT and Gapformer (see line 214)**, we use GOAT as the base model for this task. To further validate the generability and effectiveness of our HDSE framework, we also experimented with another linear transformer model, Gapformer, and observed promising results, as reported below.
>
>
>
> |                      | Cora↑          | CiteSeer↑      | PubMed↑        |
> | -------------------- | -------------- | -------------- | -------------- |
> | Gapformer            | 87.3 ± 0.7     | 76.2 ± 1.4     | 88.9 ± 0.4     |
> | **Gapformer + HDSE** | **88.4 ± 0.7** | **76.9 ± 0.6** | **89.7 ± 0.5** |
>
> Please note that we followed the supervised split setting (48%/32%/20% training/validation/test sets) used in the Gapformer paper. We are committed to conduct more experiments to further substantiate our findings and incorporate the results in the revised version.
>
> >  The results of node classification on classical datasets Core, CiteSeer and PubMed should be reported and discussed in the Evaluation section rather than Appendix.
>
> We appreciate your suggestion. We placed the experiments on Cora, CiteSeer and PubMed in the Appendix due to space constraints, we will move them to the main paper given additional space in the final version.
>
> We will incorporate your feedback and the rebuttal discussion into the paper. Thank you once again for helping us improve our work.

---

> > ### Comment · Reviewer_p468 · 2024-08-12
> >
> > Thanks for your feedback. I think my concerns are addressed and I will maintain my positive score.

---

> > > ### Author Response · Authors · 2024-08-13
> > >
> > > Dear Reviewer p468,
> > >
> > > Thank you for reaffirming your rating! We greatly appreciate your positive feedback and insights.
> > >
> > > Best regards,
> > >
> > > Authors

---

### Author Rebuttal · Authors · 2024-08-07

**We express our gratitude to all reviewers for their invaluable time, effort, and the comprehensive, constructive feedback they have provided!**

--------

**G1 Additional Visualizations**

We attach a one-page PDF that contains additional visualization results as suggested by the Reviewer 4HNX. In Figure 4 of Appendix D, the node used for visualization was selected randomly. We have clearly marked this node in the **attached PDF** (Figure 1 in PDF) and will update the manuscript accordingly. Furthermore, we have conducted additional visualizations on other randomly selected nodes, also included in the **attached PDF** (Figure 1 in PDF). These visualizations confirm that different selected nodes lead to different attention weights and consistently demonstrate our HDSE's capability to capture a multi-level hierarchical structure.

--------

In the following separate responses, we address each weakness and question raised and will incorporate the suggestions and new results into our revised paper. We are happy to provide additional information if needed.

---

### Decision · Program_Chairs · 2024-09-25

**Decision:**

Accept (poster)

**Comment:**

This manuscript introduces a hierarchical distance structural encoding (HDSE) method to enhance the performance of graph transformers by incorporating hierarchical structural information. Theoretical analysis demonstrates that the HDSE-equipped graph transformer improves expressiveness and generalization, while empirical experiments validate its effectiveness and efficiency. The proposed method outperforms traditional shortest-path structural encoding, offering superior results in both node-level and graph-level tasks.

The reviewers unanimously recommend that this manuscript be accepted. However, they also suggest including a more comprehensive ablation study and caution against making overclaims. The authors are advised to incorporate the reviewers’ feedback when preparing the final version of the paper.